# COMETH: A CONTINUOUS-TIME DISCRETE-STATE GRAPH DIFFUSION MODEL ✎

## ABSTRACT

Discrete-state denoising diffusion models led to state-of-the-art performance in *graph generation*, especially in the molecular domain. Recently, they have been transposed to continuous time, allowing more flexibility in the reverse process and a better trade-off between sampling efficiency and quality, though they have not yet been applied to graphs. Here, to leverage the benefits of both approaches, we propose COMETH, a continuous-time discrete-state graph diffusion model, tailored to the specificities of graph data. In addition, we also successfully replaced the set of structural features previously used in discrete graph diffusion models with a single random-walk-based encoding, providing a simple and principled way to boost the model's expressive power. Empirically, we show that integrating continuous time leads to significant improvements across various metrics over state-of-the-art discrete-state diffusion models on a large set of molecular and non-molecular benchmark datasets. In terms of valid, unique, and novel (VUN) samples, COMETH obtains a near-perfect performance of $99.5\%$ on the planar graph dataset and outperforms DIGRESS by $12.6\%$ on the large GuacaMol dataset.

## 1 INTRODUCTION

*Denoising diffusion models* (Ho et al., 2020; Song et al., 2020) are among the most prominent and successful classes of generative models. Intuitively, these models aim to denoise diffusion trajectories and produce new samples by sampling noise and recursively denoising it, often outperforming competing architectures in tasks such as image and video generation (Sohl-Dickstein et al., 2015; Yang et al., 2023). Recently, a large set of works, e.g., Chen et al. (2023); Jo et al. (2022; 2024); Vignac et al. (2022), aimed to leverage diffusion models for *graph generation*, e.g., the generation of molecular structures. One class of such models embeds the graphs into a *continuous space* and adds Gaussian noise to the node features and graph adjacency matrix (Jo et al., 2022). However, such noise destroys the graph's sparsity, resulting in complete, noisy graphs without meaningful structural information, making it difficult for the denoising network to capture the structural properties of the data. Therefore, *discrete-state* graph diffusion models such as DIGRESS (Vignac et al., 2022) have been proposed, exhibiting competitive performance against their continuous-state counterparts. These models utilize a categorical corruption process (Austin et al., 2021), making them more suited to the discrete structure of graph data.

In parallel, the above Gaussian-noise-based diffusion models have been extended to *continuous time* (Song et al., 2020), i.e., relying on a continuous-time stochastic process (Capasso and Bakstein, 2021), by formulating the forward process as a stochastic differential equation. In addition, discrete-state diffusion models have recently been transposed to continuous time (Campbell et al., 2022; Sun et al., 2022), relying on *continuous-time Markov chains* (CTMC). Unlike their discrete-time counterparts, which define a fixed time scale during training, they allow training using a continuous-time scale and leave the choice of the time discretization strategy for the sampling stage. Hence, incorporating continuous time enables a more optimal balance between sampling efficiency and quality while providing greater flexibility in designing the reverse process, as various CTMC simulation tools can be utilized Campbell et al. (2022); Sun et al. (2022). However, extending continuous-time discrete-state diffusion models to graphs is not straightforward. Unlike other discrete data, such as text, where all tokens share the same support, nodes and edges in graphs have distinct attributes and must be handled separately. Furthermore, the noise models used for other data modalities may be suboptimal for graphs (Vignac et al., 2022).

**Present work** Hence, to leverage the benefits of both approaches for graph generation, i.e., discrete-state and continuous-time, we propose COMETH, a *continuous-time discrete-state graph* diffusion model,

integrating graph data into a continuous diffusion model framework; see Figure 1 for an overview of COMETH. Concretely, we

1. propose a new noise model adapted to graph specificities, featuring distinct noising processes for nodes and edges, and we extend the marginal transitions previously proposed for graph data to the continuous-time setting.

2. In addition, we successfully replace the set of structural features used in most previous discrete graph diffusion models with a random-walk-based encoding. We prove that it generalizes most of the features used in DIGRESS, hence representing a simple and elegant way to boost the model's expressivity and reach state-of-the-art performance.

3. Empirically, we show that integrating continuous-time into a discrete-state graph diffusion model leads to state-of-the-art results on synthetic and established molecular benchmark datasets across various metrics. For example, in terms of VUN samples, COMETH obtains a near-perfect performance of 99.5% on the planar graph dataset and outperforms DIGRESS by 12.6% on the large GuacaMol dataset.

*Overall, COMETH is the first graph diffusion model allowing the benefits of using a discrete-state space and the flexibility of a continuous-time scale in the design of the sampling algorithm.*

**Related work** Diffusion models are a prominent class of generative models successfully applied to many data modalities, such as images, videos, or point clouds (Yang et al., 2023).

Graph generation is a well-studied task applied to various application domains, such as molecule generation, floorplan generation, or abstract syntax tree generation (Shabani et al., 2023; Shi et al., 2019). We can roughly categorize graph generation approaches into *auto-regressive models* such as Kong et al. (2023); You et al. (2018); Zhao et al. (2024); Jang et al. (2024b) and *one-shot models* such as diffusion models. The main advantage of one-shot models over auto-regressive ones is that they generate the whole graph in a single step and do not require any complex procedure to select a node ordering. On the other hand, auto-regressive models are more flexible regarding the size of the generated graph, which can remain unknown beforehand, and they do not suffer from the quadratic complexity of one-shot models.

While the first diffusion models for graph generation leveraged continuous-state spaces (Niu et al., 2020), they are now largely replaced by discrete-state diffusion models (Haefeli et al., 2022; Vignac et al., 2022), using a discrete-time scale. However, using discrete-time constrains the sampling scheme to a particular form called *ancestral sampling*, which prevents the exploration of alternative sampling strategies that could optimize sampling time or enhance sampling quality.

Another line of research considers lifting the graphs into a continuous-state space and applying Gaussian noise to the node and edge features matrices (Niu et al., 2020; Jo et al., 2022; 2024). Such continuous noise allows the generation of continuous features to be handled smoothly, such as the generation of atomic coordinates in molecular graphs (Jo et al., 2024). The above Gaussian-noise-based diffusion models have many successful applications in computational biology (Corso et al. (2023); Yim et al. (2023)). However, they almost exclusively consider point-cloud generation, focusing on modeling the geometry of the molecules and ignoring structural information. In addition, some hybrid approaches also exist that consider jointly modeling the 2D molecular graphs and their 3D geometry (Hua et al., 2024; Le et al., 2023; Vignac et al., 2023). These models usually rely on continuous noise for the atomic coordinates and discrete noise for the atom and edge types.

Recent works have tried to scale graph generative models in size (Bergmeister et al., 2023; Luo et al., 2023; Qin et al., 2023). Such frameworks are often built on top of previously proposed approaches, e.g., SPARSEDIFF (Qin et al., 2023) is based on DIGRESS. Therefore, these scaling methods are likely to apply to our approach.

Concurrently with our work, Xu et al. (2024) proposed a continuous-time discrete-state graph diffusion model. While they experiment using an MPNN as the backbone with limited success, we effectively utilize RRWP as a positional encoding. Moreover, our experimental evaluation of COMETH is more comprehensive, as we successfully implement the predictor-corrector mechanism and include a conditional generation experiment.

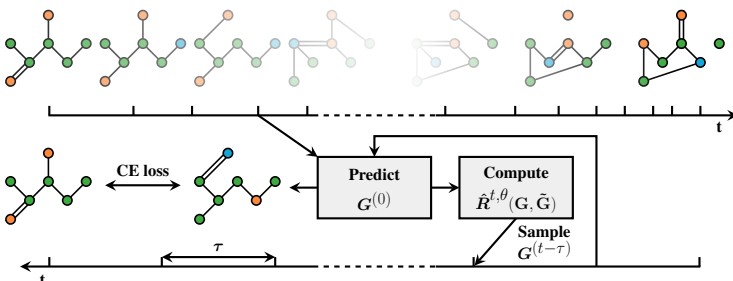

Figure 1: Overview of COMETH. During training, the graph transformer learns to predict the clean graph $\boldsymbol{G}^{(0)}$ from a noisy graph $\boldsymbol{G}^{(t)}$. Unlike previous discrete-time diffusion models, COMETH performs transitions at any time $t \in [0, 1]$. During sampling, the clean graph $\boldsymbol{G}^{(0)}$ is first predicted and used to compute the reverse rate $\hat{R}^{t,\theta}(\boldsymbol{G}, \tilde{\boldsymbol{G}})$ as defined in eq. (2). Next, a $\tau$-leaping step is performed to sample $\boldsymbol{G}^{(t-\tau)}$, with the step length fixed to $\tau$. Optionally, multiple corrector steps can be applied at $t - \tau$, which experimentally improves sample quality.

## 2 BACKGROUND

In the following, we overview the continuous-time discrete-state diffusion framework on which COMETH builds. *We provide a complete description of this framework in Appendix A.1*, providing intuitions and technical details, and refer to Yang et al. (2023) for a general introduction to diffusion models.

**Continuous-time discrete diffusion** In the discrete diffusion setting, we aim at modeling a discrete data distribution $p_{\text{data}}(z^{(0)})$, where $z^{(0)} \in \mathcal{Z}$ and $\mathcal{Z}$ is a finite set with cardinality $S := |\mathcal{Z}|$. A *continuous-time diffusion model* (Campbell et al., 2022; Sun et al., 2022; Lou et al., 2023) is a *stochastic process*, running from time $t = 0$ to $t = 1$. In the following, we denote the marginal distributions of the state $z^{(t)} \in \mathcal{Z}$ at time $t$ by $q_t(z^{(t)})$, and $q_{t|s}(z^{(t)} \mid z^{(s)})$ denotes the conditional distribution of the state $z^{(t)}$ given the state $z^{(s)} \in \mathcal{Z}$ at some time $s \in [0, 1]$. We also denote $\delta_{\tilde{z},z}$ the Kronecker delta, which is equal to 1 if $\tilde{z} = z$ and 0 otherwise. We define a *forward process* which gradually transforms the marginal distribution $q_0(z^{(0)}) = p_{\text{data}}(z^{(0)})$ into $q_1(z^{(1)})$, that is "close" to an easy-to-sample prior distribution $p_{\text{ref}}(z^{(1)})$, e.g., a uniform categorical distribution.

We define the forward process as a *continuous-time Markov chain* (CTMC). The current state $z^{(t)}$ alternates between resting in the current state and transitioning to another state, where a *transition rate matrix* $\boldsymbol{R}^{(t)} \in \mathbb{R}^{S \times S}$ controls the dynamics of the CTMC. Formally, the *forward process* is defined through the infinitesimal transition probability from $z^{(t)}$ to another state $\tilde{z} \in \mathcal{Z}$, for a infinitesimal time step $dt$ between time $t$ and $t + dt$,

$$q_{t+dt|t}\left(\tilde{z} \mid z^{(t)}\right) := \delta_{\tilde{z},z^{(t)}} + R^{(t)}\left(z^{(t)}, \tilde{z}\right)dt,$$

where $R^{(t)}(z^{(t)}, \tilde{z})$ denotes the entry of $\boldsymbol{R}^{(t)}$ that gives the rate from $z^{(t)}$ to $\tilde{z}$. Intuitively, the process is more likely to transition to a state where $R^{(t)}(z^{(t)}, \tilde{z})$ is high, and $\boldsymbol{R}^{(t)}$ is designed so that $q_1(z^{(1)})$ closely approximates the prior distribution $p_{\text{ref}}$.

The generative process is formulated as the time reversal of the forward process, therefore interpolating from $q_1(z^{(1)})$ to $p_{\text{data}}(z^{(0)})$. The rate of this reverse CTMC, $\hat{\boldsymbol{R}}^{(t)}$, is intractable (Campbell et al., 2022) and has to be modeled by a parametrized approximation, i.e.,

$$\hat{R}^{t,\theta}(z, \tilde{z}) = R^{(t)}(\tilde{z}, z) \sum_{z^{(0)} \in \mathcal{Z}} \frac{q_{t|0}\left(\tilde{z} \mid z^{(0)}\right)}{q_{t|0}\left(z \mid z^{(0)}\right)} p_{0|t}^{\theta}\left(z^{(0)} \mid z\right), \text{ for } z \neq \tilde{z},$$

where $p_{0|t}^{\theta}(z^{(0)} \mid z)$ is the *denoising neural network* with parameters $\theta$.

For efficient training, the conditional distribution $q_{t|0}(z^{(t)} \mid z^{(0)})$ needs to be analytically obtained; see Appendix A.1 for more details. As demonstrated in Campbell et al. (2022), this property is achieved

by choosing $\boldsymbol{R}^{(t)} = \beta(t)\boldsymbol{R}_b$ with $\beta(t) \in \mathbb{R}$ being the *noise schedule* and $\boldsymbol{R}_b \in \mathbb{R}^{S \times S}$ is a constant base rate matrix. We can now generate samples by simulating the reverse process from $t = 1$ to $t = 0$. Different algorithms can be employed for this purpose, such as Euler's method (Sun et al., 2022) or $\tau$-leaping (Campbell et al., 2022).

# 3 A Continuous-Time Discrete-State Graph Diffusion Model

Here, we present our COMETH framework, a continuous-time discrete-state diffusion model for graph generation. Let $[\![m, n]\!] := \{m, \ldots, n\} \subset \mathbb{N}$. We denote $n$-order attributed graph as a pair $\boldsymbol{G} := (G, \boldsymbol{X}, \mathbf{E})$, where $G := (V(G), E(G))$ is a graph, $\boldsymbol{X} \in \{0, 1\}^{n \times a}$, for $a > 0$, is a *node feature matrix*, and $\mathbf{E} \in \{0, 1\}^{n \times n \times b}$, for $b > 0$, is an *edge feature tensor*. Note that node and edge features are considered to be discrete and consequently encoded in a one-hot encoding. For notational convenience, in the following, we denote the graph at time $t \in [0, 1]$ by $\boldsymbol{G}^{(t)}$, the node feature of node $i \in V(G)$ at time $t$ by $x_i^{(t)} \in [\![1, a]\!]$, and similarly the edge feature of edge $(i, j) \in E(G)$ at time $t$ by $e_{ij}^{(t)} \in [\![1, b]\!]$. In addition, we treat the absence of an edge as a special edge with a unique edge feature. By $\mathbb{1}$, we denote an all-one vector of appropriate size, by $\boldsymbol{I}$, the identity matrix of appropriate size, while $\mathbf{o}$ denotes the all-zero matrix of appropriate size. Moreover, by $\boldsymbol{a}'$, we denote the transpose of the vector $\boldsymbol{a}$.

## 3.1 Forward process factorization

Considering the graph state-space would result in a prohibitively large state, making it impossible to construct a transition matrix. Therefore, we consider that the forward process factorizes and that the noise propagates independently on each node and edge, enabling us to consider node and edge state spaces separately. Formally, let $\boldsymbol{G} = (G, \boldsymbol{X}, \mathbf{E})$ be $n$-order attributed graph, then we have

$$q_{t+dt|t}\left(\boldsymbol{G}^{(t+dt)} \mid \boldsymbol{G}^{(t)}\right) := \prod_{i=1}^{n} q_{t+dt|t}\left(x_i^{(t+dt)} \mid x_i^{(t)}\right) \prod_{i<j}^{n} q_{t+dt|t}\left(e_{ij}^{(t+dt)} \mid e_{ij}^{(t)}\right).$$

The above factorization reveals a challenge not yet addressed for the continuous-time diffusion model. In other types of discrete data, such as text, all tokens share the same support. In contrast, nodes and edges have different attributes, and their respective sets of labels may have different sizes. We, therefore, need to define their respective forward processes differently.

We then define a pair of rate matrices $(\boldsymbol{R}_X^{(t)}, \boldsymbol{R}_E^{(t)})$, with $\boldsymbol{R}_X^{(t)} := \beta(t)\boldsymbol{R}_X$ and $\boldsymbol{R}_E^{(t)} := \beta(t)\boldsymbol{R}_E$, where $\beta$ is the noise schedule and $\boldsymbol{R}_X \in \mathbb{R}^{d \times d}$, $\boldsymbol{R}_E \in \mathbb{R}^{e \times e}$ are base rate matrices for nodes and edges, respectively. The two matrices differ in size and allow for controlling the dynamics of the forward process in distinct ways for nodes and edges. Note that we followed the design choice of Campbell et al. (2022) introduced in Section 2.

## 3.2 Noise model : extending marginal transitions to continuous time

**Noise model** Several noise models have been proposed for discrete diffusion models, including uniform transitions, absorbing transitions (Austin et al., 2021), and marginal transitions (Vignac et al., 2022). We propose to use a rate matrix that is analogous to the marginal transition matrix, as it is well adapted for graph data. However, marginal transitions have not been utilized in the continuous-time framework. To address this, we extend this concept by constructing the following rate matrices, i.e.,

$$\boldsymbol{R}_X = \mathbb{1}\boldsymbol{m}_X' - \boldsymbol{I} \quad \text{and} \quad \boldsymbol{R}_E = \mathbb{1}\boldsymbol{m}_E' - \boldsymbol{I},$$

where $\boldsymbol{m}_X$ and $\boldsymbol{m}_E$ are vectors representing the marginal distributions $m_X$ and $m_E$ of node and edge labels, i.e., they contain the frequency of the different labels in the dataset. With such a rate, the transition rate to a particular state depends on its marginal probability. Specifically, the more common a node or edge label is in the dataset, the higher the transition rate to that label. Consequently, this approach helps preserve sparsity in noisy graphs by favoring transitions to the "no edge" label. Embedding this inductive bias in the noise model simplifies the model's task, as it no longer needs to reconstruct this sparsity during the generative process.

**Deriving an explicit expression for the forward process** We now aim to understand the relationship between the rate matrix and the endpoint of the diffusion process at $t = 1$, $q_1(\boldsymbol{G}^{(1)})$. In fact, we need to

choose $p_{\text{ref}}$ so that $q_1(\boldsymbol{G}) \approx p_{\text{ref}}(\boldsymbol{G})$ for determining the appropriate prior distribution that aligns with the chosen rate matrix. Ideally, we seek to use the product of distributions $\prod_i^n m_X \prod_{i<j}^n m_E$, which, as demonstrated in Vignac et al. (2022, Theorem 4.1), is optimal within the space of distributions factorized over nodes and edges. In the following, we explain how to design the noise schedule $\beta$ to achieve this.

To better understand the relationship between the rate matrix and $q_1(z^{(1)})$, we require an explicit expression that readily links $q_{t|0}(z^{(t)} \mid z^{(0)})$ to $\boldsymbol{R}^{(t)}$. However, Campbell et al. (2022) provide the following closed-form for the former, which does not easily allow for the direct deduction of an appropriate prior distribution,

$$q_{t|0}\Big(z^{(t)} = k \mid z^{(0)} = l\Big) = \left(\boldsymbol{P} \exp\left[\boldsymbol{\Lambda} \int_0^t \beta(s)ds\right] \boldsymbol{P}^{-1}\right)_{kl}, \tag{1}$$

where $\boldsymbol{R}_b = \boldsymbol{P}\boldsymbol{\Lambda}\boldsymbol{P}^{-1}$ and $\exp$ refers to the element-wise exponential. Given Equation (1), it is not straightforward to determine the form of $q_{1|0}(z^{(1)} \mid z^{(0)})$, and, consequently, the form of $q_1(z^{(1)})$.

We therefore prove that this expression can be refined, offering clearer insights into the behavior of the forward process.

**Proposition 1.** For a CTMC $(z^{(t)})_{t\in[0,1]}$ with rate matrix $\boldsymbol{R}^{(t)} = \beta(t)\boldsymbol{R}_b$ and $\boldsymbol{R}_b = \mathbb{1}\boldsymbol{m}' - \boldsymbol{I}$, the forward process can be written as

$$\bar{\boldsymbol{Q}}^{(t)} = e^{-\bar{\beta}^{(t)}}\boldsymbol{I} + \left(1 - e^{-\bar{\beta}^{(t)}}\right)\mathbb{1}\boldsymbol{m}',$$

where $(\bar{\boldsymbol{Q}}^{(t)})_{ij} = q(z^{(t)} = i \mid z^0 = j)$ and $\bar{\beta}^{(t)} = \int_0^t \beta(s)ds$.

Therefore, if we can design $\bar{\beta}^{(t)}$ so that $\lim_{t\to 1} e^{-\bar{\beta}^{(t)}} = 0$, we get that $\lim_{t\to 1} \bar{\boldsymbol{Q}}^{(t)} = \mathbb{1}\boldsymbol{m}'$, which means that $z^{(1)}$ is sampled from the categorical distribution $m$ whatever the value of $z^{(0)}$, i.e., $q_1(z^{(1)}) = m$. In our case, since Proposition 1 holds for every node and edge, and given that the forward process is factorized, this would yield that $q_1(\boldsymbol{G}^{(1)}) = \prod_i^n m_X \prod_{i<j}^n m_E$ as desired. We provide a more detailed explanation in Appendix A.2.

**Proposed noise schedule** Even though, in theory, one should set $\lim_{t\to 1} \bar{\beta}^{(t)} = +\infty$ so that $\lim_{t\to 1} \bar{\boldsymbol{Q}}^{(t)} = \mathbb{1}\boldsymbol{m}'$, it considerably restricts the space of possible noise schedules. Relying on the exponentially decreasing behavior of the cumulative noise schedule $e^{-\bar{\beta}^{(t)}}$, one only needs to ensure that $\bar{\beta}^{(1)}$ is high enough so that $\bar{\boldsymbol{Q}}^{(1)}$ satisfyingly approximate $\mathbb{1}\boldsymbol{m}'$. Campbell et al. (2022) proposed an exponential noise schedule for categorical data. Instead, we rather followed an older heuristic, i.e. using a cosine-shaped schedule as introduced for discrete-time models in Nichol and Dhariwal (2021). We therefore propose to use a *cosine noise schedule*, where

$$\beta(t) = \alpha\frac{\pi}{2}\sin\left(\frac{\pi}{2}t\right) \quad \text{and} \quad \int_0^t \beta(s)ds = \alpha\left(1 - \cos\left(\frac{\pi}{2}t\right)\right).$$

Here, $\alpha$ is a constant factor. Since $e^{-\bar{\beta}^1} = e^{-\alpha}$, given that $\alpha$ is high enough, $q_1(\boldsymbol{G}^{(1)})$ will be close to $\mathbb{1}\boldsymbol{m}'$, and we can therefore use $\prod_i^n m_X \prod_{i<j}^n m_E$ as our prior distribution. We provide more intuition on our noise schedule in Appendix B.3. Finally, noising an $n$-order attributed graph $\boldsymbol{G} = (G, \boldsymbol{X}, \mathbf{E})$ amounts to sample from the following distribution,

$$q_{t|0}\Big(\boldsymbol{G}^{(t)} \mid \boldsymbol{G}\Big) = \left(\boldsymbol{X}\bar{\boldsymbol{Q}}_X^{(t)}, \mathbf{E}\bar{\boldsymbol{Q}}_E^{(t)}\right).$$

Since we consider only undirected graphs, we apply noise to the upper-triangular part of $\mathbf{E}$ and symmetrize. Note that we apply noise to a graph in the same manner as in discrete-time models, with the only difference being that $t$ is no longer discrete.

## 3.3 PARAMETRIZATION AND OPTIMIZATION

We can formulate the approximate reverse rate for our graph generation model. We set the unidimensional rates according to the parametrized approximation derived by Campbell et al. (2022):

$$\hat{R}_X^{t,\theta}\Big(x_i^{(t)}, \tilde{x}_i\Big) = R_X^{(t)}\Big(\tilde{x}_i, x_i^{(t)}\Big) \sum_{x_0} \frac{q_{t|0}\Big(\tilde{x}_i \mid x_i^{(0)}\Big)}{q_{t|0}\Big(x_i^{(t)} \mid x_i^{(0)}\Big)} p_{0|t}^\theta\Big(x_i^{(0)} \mid \boldsymbol{G}^{(t)}\Big), \text{ for } x_i^{(t)} \neq \tilde{x}_i, \tag{2}$$

and similarly for the edge reverse rate $\hat{R}_E^{t,\theta}\left(e_{ij}^{(t)}, \tilde{e}_{ij}\right)$. We elaborate on how to use those rates to simulate the reverse process in Section B.1. Formally, we set the overall reverse rate to

$$\hat{R}^{t,\theta}(\boldsymbol{G}, \tilde{\boldsymbol{G}}) = \sum_i \delta_{\boldsymbol{G}^{\backslash x_i}, \tilde{\boldsymbol{G}}^{\backslash x_i}} \hat{R}_X^{t,\theta}(x_i^{(t)}, \tilde{x}) + \sum_{i<j} \delta_{\boldsymbol{G}^{\backslash e_{ij}}, \tilde{\boldsymbol{G}}^{\backslash e_{ij}}} \hat{R}_E^{t,\theta}(e_{ij}^{(t)}, \tilde{e}_{ij}),$$

where $\delta_{\boldsymbol{G}^{\backslash x_i}, \tilde{\boldsymbol{G}}^{\backslash x_i}}$ is 1 if $\boldsymbol{G}$ and $\tilde{\boldsymbol{G}}$ differ only on node $i$ and 0 otherwise, and similarly $\delta_{\boldsymbol{G}^{\backslash e_{ij}}, \tilde{\boldsymbol{G}}^{\backslash e_{ij}}}$ for edges. The reverse rate exhibits the classic diffusion model parametrization, which relies on predicting a clean data point $\boldsymbol{G}$ given a noisy input $\boldsymbol{G}^{(t)}$. We, therefore, train a denoising neural network $p_{0|t}^\theta(\boldsymbol{G} \mid \boldsymbol{G}^{(t)})$ to this purpose. The outputs are normalized into probability distributions for node and edge labels.

The model can be optimized using the continuous-time ELBO proposed in Campbell et al. (2022). Additionally, they incorporate direct model supervision by optimizing an auxiliary objective, i.e. the cross-entropy loss between the predicted clean graph and the ground truth $\boldsymbol{G}^{(0)}$. However, our preliminary experiments with this ELBO yielded poor empirical results, as detailed in appendix D.4. Given that the ELBO and cross-entropy share the same optimum, and the cross-entropy loss alone has been successfully applied for graphs in the discrete-time case (Vignac et al., 2022), we opted to use the cross-entropy loss $\mathcal{L}_{\text{CE}}$ as our optimization objective:

$$\mathbb{E}_{t \sim \mathcal{U}([0,1]), p_{\text{data}}(\boldsymbol{G}^{(0)}), q(\boldsymbol{G}^{(t)} | \boldsymbol{G}^{(0)})} \left[ -\sum_i^n \log p_{0|t}^\theta\left(x_i^{(0)} \mid \boldsymbol{G}^{(t)}\right) - \lambda \sum_{i<j}^n \log p_{0|t}^\theta\left(e_{ij}^{(0)} \mid \boldsymbol{G}^{(t)}\right) \right], \quad (3)$$

where $\lambda \in \mathbb{R}^+$ is a scaling factor that controls the relative importance of edges and nodes in the loss. In practice, we set $\lambda > 1$ so that the model prioritizes predicting a correct graph structure over predicting correct node labels.

## 3.4 SIMPLE AND POWERFUL POSITIONAL ENCODING WITH RRWP

In all our experiments, we use the graph transformer proposed by Vignac et al. (2022); see Figure 4 in the appendix. Relying on the fact that discrete diffusion models preserve the sparsity of noisy graphs, they propose a large set of features to compute at each denoising step to boost the expressivity of the model. This set includes Laplacian features, connected components features, and node- and graph-level cycle counts. Even though this set of features has been successfully used in follow works, e.g., Vignac et al. (2023), Qin et al. (2023), Igashov et al. (2023), no theoretical nor experimental study exists to investigate the relevance of those particular features. In addition, a rich literature on encodings in graph learning has been developed, e.g., LapPE (Kreuzer et al., 2021), SignNet (Lim et al., 2023), RWSE (Dwivedi et al., 2021), SPE (Huang et al., 2023), which led us to believe that powerful encodings developed for discriminative models should be transferred to the generative setting.

Specifically, in our experiments, we leverage the *relative random-walk probabilites* (RRWP) encoding, introduced in Ma et al. (2023). Denoting $\boldsymbol{A}$ the adjacency matrix of a graph $G$, $\boldsymbol{D}$ the diagonal degree matrix, and $\boldsymbol{M} = \boldsymbol{D}^{-1}\boldsymbol{A}$ the degree-normalized adjacency matrix, for each pair of nodes $(i, j) \in V(G)^2$, the RRWP encoding computes

$$P_{ij}^K := \left[ I_{ij}, M_{ij}, M_{ij}^2, \ldots, M_{ij}^{K-1} \right], \quad (4)$$

where $K$ refers to the maximum length of the random walks. The entry $P_{ii}^K$ corresponds to the RWSE encoding of node $i$; therefore, we leverage them as node encodings. This encoding provides an efficient and elegant solution for boosting model expressivity and performance through a unified encoding.

In the following, we show that RRWP encoding can (approximately) determine if two nodes lie in the same connected components, approximate the size of the largest connected component, and count small cycles.

**Theorem 2 (Informal).** For $n \in \mathbb{N}$, let $\mathcal{G}_n$ denote the set of $n$-order graphs and for a graph $G \in \mathcal{G}_n$ let $V(G) := [\![1, n]\!]$. Then RRWP composed with a universally approximating feed-forward neural network can

1. determine if two vertices are in the same connected component,

2. approximate the size of the largest connected component in $G$,

3. approximately count the number $p$-cycles, for $p < 5$, in which a node is contained.

See Appendix A.3 for the detailed formal statements. However, we can also show that RRWP encodings cannot detect if a node is contained in a large cycle of a given graph. We say the RRWP encoding counts the number of $p$-cycles for $p \geq 2$ if there do not exist two graphs, one containing at least one $p$-cycle while the other does not, while the RRWP encodings of the two graphs are equivalent.

**Proposition 3.** For $p \geq 8$, the RRWP encoding does not count the number of $p$-cycles.

Hence, for $p \geq 8$ and $K \geq 0$, there exists a graph and two vertex pairs $(r, s), (v, w) \in V(G)^2$ such that $(r, s)$ is contained in $C$ while $(v, w)$ is not and $P_{vw}^K = P_{rs}^K$.

**Equivariance properties** Since graphs are invariant to node reordering, it is essential to design methods that capture this fundamental property of the data. Relying on the similarities between COMETH and DIGRESS, we establish that COMETH is permutation-equivariant and that our loss is permutation-invariant. We also establish that the $\tau$-leaping sampling scheme and the predictor-corrector yield exchangeable distributions, i.e., the model assigns each graph permutation the same probability. Since those results mainly stem from proofs outlined in Vignac et al. (2022), we moved them to Appendix A.4.

## 4 EXPERIMENTS

We empirically evaluate COMETH on synthetic and real-world graph generation benchmarks in the following. For all datasets, the results obtained with the raw model are denoted as COMETH, while the results using the predictor-corrector are referred to as COMETH-PC. We sample from Cometh using the tau-leaping algorithm as described in Campbell et al. (2022), a procedure that we detail in Appendix B.1. We also conduct several ablation studies in Appendix D. Finally, we describe our conditional generation setting in appendix B.2. The code will be made publicly available in the near future to ensure reproducibility.

### 4.1 SYNTHETIC GRAPH GENERATION: PLANAR AND SBM

Here, we outline experiments regarding synthetic graph generation.

**Datasets and metrics** We first validate COMETH on two benchmarks proposed by Martinkus et al. (2022), PLANAR and SBM. We measure four standard metrics to assess the ability of our model to capture structural properties of the graph distributions, i.e., degree (**Degree**), clustering coefficient (**Cluster**), orbit count(**Orbit**), and the eigenvalues of the graph Laplacian (**Spectrum**). The reported values are the maximum mean discrepancies (MMD) between those metrics evaluated on the generated graphs and the test set. We also report the percentage of valid, unique, and novel (**VUN**) samples among the generated graphs to further assess the ability of our model to capture the properties of the targeted distributions correctly. We provide a detailed description of the metrics in Appendix C.1.

**Baselines** We evaluate our model against several graph diffusion models, namely DIGRESS (Vignac et al., 2022), GRUM (Jo et al., 2024), two autoregressive models, GRAN (Liao et al., 2019), and GRAPHRNN (You et al., 2018), and a GAN, SPECTRE (Martinkus et al., 2022). We re-evaluated previous state-of-the-art models over five runs, namely DIGRESS and GRUM, to provide error bars for the results.

**Results** See Table 1. On PLANAR, our COMETH yields very good results over all metrics, being only outperformed by DIGRESS on **Degree** and **Orbit**, but with a much lower **VUN**. We observe that the sampling quality benefits from the predictor-corrector scheme, with a near-perfect **VUN** score. On SBM, we also obtain state-of-the-art results on all metrics. However, we found that the predictor-corrector did not improve performance on this dataset.

### 4.2 SMALL-MOLECULE GENERATION: QM9

Here, we outline experiments regarding small-molecule generation.

**Datasets and metrics** To assess the ability of our method to model attributed graph distributions, we evaluate its performance on the standard dataset QM9 (Wu et al. (2018)). Molecules are kekulized using the RDKit library, removing hydrogen atoms. We use the same split as Vignac et al. (2022), with 100k

Table 1: **Synthetic graph generation results.** We report the mean of five runs, as well as 95% confidence intervals. We reproduced the baselines results for DiGRESS and GRuM, the rest is taken from Jo et al. (2024). Best results are highlighted in bold.

| Model | Degree ↓ | Cluster ↓ | Orbit ↓ | Spectrum ↓ | VUN [%] ↑ |
|---|---|---|---|---|---|
| *Planar graphs* | | | | | |
| GRAPHRNN | 24.5 | 9.0 | 2508 | 8.8 | 0 |
| GRAN | 3.5 | 1.4 | 1.8 | 1.4 | 0 |
| SPECTRE | 2.5 | 2.5 | 2.4 | 2.1 | 25 |
| DiGRESS | $0.8_{\pm0.0}$ | $4.1_{\pm0.3}$ | $\mathbf{0.5}_{\pm0.0}$ | – | $76.0_{\pm0.1}$ |
| GRuM | $2.6_{\pm1.7}$ | $1.3_{\pm0.3}$ | $10.0_{\pm7.7}$ | $1.4_{\pm0.2}$ | $91.0_{\pm5.7}$ |
| DISCO | $\mathbf{1.2}_{\pm0.5}$ | $1.3_{\pm0.5}$ | $1.7_{\pm0.7}$ | - | $83.6_{\pm2.1}$ |
| COMETH | $2.1_{\pm1.3}$ | $1.5_{\pm0.4}$ | $3.1_{\pm3.0}$ | $\mathbf{1.3}_{\pm0.3}$ | $92.5_{\pm3.7}$ |
| COMETH–PC | $2.0_{\pm0.9}$ | $\mathbf{1.1}_{\pm0.1}$ | $7.7_{\pm3.8}$ | $\mathbf{1.3}_{\pm0.2}$ | $\mathbf{99.5}_{\pm0.9}$ |
| *Stochastic block model* | | | | | |
| GRAPHRNN | 6.9 | 1.7 | 3.1 | 1.0 | 5 |
| GRAN | 14.1 | 1.7 | 2.1 | 0.9 | 25 |
| SPECTRE | 1.9 | 1.6 | 1.6 | 0.9 | 53 |
| DiGRESS | $1.7_{\pm0.1}$ | $5.0_{\pm0.1}$ | $3.6_{\pm0.4}$ | – | $74.0_{\pm4.0}$ |
| GRuM | $2.6_{\pm1.0}$ | $1.5_{\pm0.0}$ | $1.8_{\pm0.4}$ | $0.9_{\pm0.2}$ | $69.0_{\pm8.5}$ |
| DISCO | $\mathbf{0.8}_{\pm0.2}$ | $\mathbf{0.8}_{\pm0.4}$ | $2.0_{\pm0.5}$ | - | $66.2_{\pm1.4}$ |
| COMETH | $2.4_{\pm1.1}$ | $1.5_{\pm0.0}$ | $\mathbf{1.7}_{\pm0.2}$ | $\mathbf{0.8}_{\pm0.1}$ | $\mathbf{77.0}_{\pm5.3}$ |

Table 2: **Molecule generation results on QM9.** We report the mean of five runs, as well as 95% confidence intervals. Best results are highlighted in bold. Baseline results are taken from Jang et al. (2024a).

| Model | Validity ↑ | Uniqueness ↑ | Valid & Unique ↑ | FCD ↓ | NSPDK ↓ |
|---|---|---|---|---|---|
| GDSS | 95.72 | $\mathbf{98.46}$ | 94.25 | 2.9 | 0.003 |
| DiGRESS | 99.01 | 96.34 | 95.39 | $\mathbf{0.25}$ | 0.001 |
| GRAPHARM | 90.25 | 95.26 | 85.97 | 1.22 | 0.002 |
| HGGT | 99.22 | 95.65 | 94.90 | 0.40 | $\mathbf{0.000}$ |
| COMETH | $\mathbf{99.57}_{\pm0.07}$ | $96.76_{\pm0.17}$ | $\mathbf{96.34}_{\pm0.2}$ | $\mathbf{0.25}_{\pm0.01}$ | $\mathbf{0.000}_{\pm0.00}$ |

molecules for training, 10k for testing, and the remaining data for the validation set. We want to stress that this split differs from Jo et al. (2022), which uses $\sim 120$k molecules for training and the rest as a test set. We choose to use the former version of this dataset because it allows for selecting the best checkpoints based on the evaluation of the ELBO on the validation set. We report the **Validity** over 10k molecules, as evaluated by RDKit sanitization, as well as the **Uniqueness**, **FCD**, using the MOSES benchmark, and **NSPDK**. Appendix C.2 provides a complete description of the metrics.

**Baselines** We evaluate our model against several recent graph generation models, including two diffusion models, DiGRESS Vignac et al. (2022) and GDSS (Jo et al., 2022)), and two autoregressive models, HGGT Jang et al. (2024a) and GRAPHARM Kong et al. (2023)).

**Results** We report results using 500 denoising steps for a fair comparison, as in Vignac et al. (2022). COMETH performs very well on this simple molecular dataset, notably outperforming its discrete-time counterpart DiGRESS in terms of valid and unique samples with similar FCD and NSPDK. We experimentally found that the predictor-corrector does not improve performance on this dataset; therefore, we do not report results using this sampling scheme.

### 4.3 MOLECULE GENERATION ON LARGE DATASETS: MOSES AND GUACAMOL

Here, we outline experiments regarding large-scale molecule generation.

**Datasets and benchmarks** We also evaluate our model on two much larger molecular datasets, MOSES (Polykovskiy et al., 2020)) and GuacaMol (Brown et al., 2019). The former is a refinement of the ZINC database and includes 1.9M molecules, with 1.6M allocated to training. The latter is derived from the ChEMBL database and comprises 1.4M molecules, from which 1.1M are used for training. We use a preprocessing step similar to Vignac et al. (2022) for the GuacaMol dataset, which filters molecules that cannot be mapped from SMILES to graph and back to SMILES.

Table 3: **Molecule generation on MOSES.** We report the mean of five runs, as well as 95% confidence intervals. Best results are highlighted in bold, and second best results are underlined.

| Model | Class | Val. ↑ | Val. & Uni. ↑ | VUN ↑ | Filters ↑ | FCD ↓ | SNN ↑ | Scaf ↑ |
|---|---|---|---|---|---|---|---|---|
| VAE | Smiles | 97.7 | 97.5 | 67.8 | **99.7** | **0.57** | **0.58** | 5.9 |
| JT-VAE | Fragment | **100** | **100** | 99.9 | 97.8 | 1.00 | 0.53 | 10 |
| GraphInvent | Autoreg. | 96.4 | 96.2 | – | 95.0 | 1.22 | 0.54 | 12.7 |
| DiGress | One-shot | 85.7 | 85.7 | 81.4 | 97.1 | 1.19 | 0.52 | 14.8 |
| Disco | One-shot | 88.3 | 88.3 | 86.3 | 95.6 | 1.44 | 0.50 | 15.1 |
| Cometh | One-shot | $87.0_{\pm 0.2}$ | $86.9_{\pm 0.2}$ | $83.8_{\pm 0.2}$ | $97.2_{\pm 0.1}$ | $1.44_{\pm 0.02}$ | $0.51_{\pm 0.0}$ | $15.9_{\pm 0.8}$ |
| Cometh–PC | One-shot | $90.5_{\pm 0.3}$ | $90.4_{\pm 0.3}$ | $83.7_{\pm 0.2}$ | $99.1_{\pm 0.1}$ | $1.27_{\pm 0.02}$ | $0.54_{\pm 0.0}$ | $16.0_{\pm 0.7}$ |

Table 4: **Molecule generation on GuacaMol.** We report the mean of five runs, as well as 95% confidence intervals. Conversely to MOSES, the GuacaMol benchmark reports scores, so higher is better. Best results are highlighted in bold, and second best results are underlined.

| Model | Class | Val.↑ | Val. & Uni.↑ | VUN↑ | KL div↑ | FCD↑ |
|---|---|---|---|---|---|---|
| LSTM | Smiles | 95.9 | 95.9 | 87.4 | **99.1** | **91.3** |
| NAGVAE | One-shot | 92.9 | 95.5 | 88.7 | 38.4 | 0.9 |
| MCTS | One-shot | **100.0** | **100.0** | 95.4 | 82.2 | 1.5 |
| DiGress | One-shot | 85.2 | 85.2 | 85.1 | 92.9 | 68.0 |
| Disco | One-shot | 86.6 | 86.6 | 86.5 | 92.6 | 59.7 |
| Cometh | One-shot | $94.4_{\pm 0.2}$ | $94.4_{\pm 0.2}$ | $93.5_{\pm 0.3}$ | $94.1_{\pm 0.4}$ | $67.4_{\pm 0.3}$ |
| Cometh–PC | One-shot | $98.9_{\pm 0.1}$ | $98.9_{\pm 0.1}$ | $97.6_{\pm 0.2}$ | $96.7_{\pm 0.2}$ | $72.7_{\pm 0.2}$ |

Both datasets come with their own metrics and baselines, which we briefly describe here. As for QM9, we report **Validity**, as well as the percentage of **Valid & Unique (Val. & Uni.)** samples, and **Valid, Unique and Novel (VUN)** samples for both datasets. We also report **Filters**, **FCD**, **SNN**, and **Scaf** for MOSES, as well as **KL div** and **FCD** for GuacaMol. These metrics are designed to evaluate the model's capability to capture the chemical properties of the learned distributions. We provide a detailed description of those metrics in Appendix C.3.

**Results** Similarly to previous graph generation models, Cometh does not match the performance of molecule generation methods that incorporate domain-specific knowledge, especially SMILES-based models (see Table 3). However, Cometh further bridges the gap between graph diffusion models and those methods, outperforming DiGress in terms of validity by a large margin.

On GuacaMol (see Table 4), Cometh obtains excellent performance in terms of VUN samples, with an impressive 12.6% improvement over DiGress. The LSTM model still surpasses our graph diffusion model on the **FCD** metric. This may be due to the fact that we train on a subset of the original dataset, whereas the LSTM is trained directly on SMILES.

## 4.4 Conditional generation

We perform conditional generation on QM9 following the setting of Vignac et al. (2022). We target two molecular properties, the **dipole moment** $\mu$ and the **highest occupied molecular orbital energy (HOMO)**. We sample 100 properties from the test set for each experiment and use them as conditioners

Table 5: **Conditional molecule generation results on QM9**. We report the mean of five runs, as well as 95% confidence intervals.

| Model | $\mu$ | | HOMO | | $\mu$ & HOMO | |
|---|---|---|---|---|---|---|
| | MAE ↓ | Val ↑ | MAE ↓ | Val ↑ | MAE ↓ | Val ↑ |
| DiGress | 0.81 | – | 0.56 | – | 0.87 | – |
| Cometh | $\mathbf{0.67}_{\pm 0.02}$ | $88.8_{\pm 0.5}$ | $\mathbf{0.32}_{\pm 0.01}$ | $94.1_{\pm 0.8}$ | $\mathbf{0.58}_{\pm 0.01}$ | $92.5_{\pm 0.7}$ |

to generate ten molecules. We estimate the properties of the sampled molecules using the Psi4 library (Smith et al. (2020)) and report the **mean absolute error (MAE)** between the estimated properties from the generated set and the targeted properties.

We report our results against DiGress in Table 5. Overall, Cometh outperforms DiGress by large margin, with 18%, 43%, and 33% improvements on $\mu$, HOMO and both targets respectively. Those performance improvements indicate the superiority of classifier-free guidance over classifier-guidance for conditional graph generation.

## 5 CONCLUSION

Here, to leverage the benefits of continuous-time and discrete-state diffusion model, we proposed COMETH, a continuous-time discrete-state graph diffusion model, integrating graph data into a continuous diffusion model framework. We introduced a new noise model adapted to graph specificities using different node and edge rates and a tailored marginal distribution and noise schedule. In addition, we successfully replaced the structural features of DIGRESS with a single encoding with provable expressivity guarantees, removing unnecessary features. Empirically, we showed that integrating continuous time leads to significant improvements across various metrics over state-of-the-art discrete-state diffusion models on a large set of molecular and non-molecular benchmark datasets.

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

# Appendices

We provide proofs and additional theoretical background in Appendix A. We give details on our implementation in Appendix B, experimental details in Appendix C and additional experimental results in Appendix D. Finally, we provide visualization for generated samples in Appendix F.

## A  THEORETICAL DETAILS

Here, we outline the theoretical details of our COMETH architecture.

### A.1  ADDITIONAL BACKGROUND

**Continuous-time discrete diffusion** Here, we provide a more detailed description of the continuous-time, discrete-state diffusion model introduced in Campbell et al. (2022).

We begin by recalling our notations. We aim to model a discrete data distribution $p_{\text{data}}(z^{(0)})$, where $z^{(0)} \in \mathcal{Z}$ and $\mathcal{Z}$ is a finite set with cardinality $S := |\mathcal{Z}|$. In the following, the state is denoted by $z^{(t)} \in \mathcal{Z}$, where time is denoted by $t \in [0,1]$, and $\boldsymbol{z}^{(t)} \in \{0,1\}^S$ is its one-hot encoding. The marginal distributions at time $t$ are denoted by $q_t(z^{(t)})$ and the conditional distribution of the state $z^{(t)}$ given the state $z^{(s)}$ at some time $s \in [0,1]$ by $q_{t|s}(z^{(t)} \mid z^{(s)})$. We also denote $\delta_{\tilde{z},z}$ the Kronecker delta, which equals 1 if $\tilde{z} = z$ and 0 otherwise.

This model builds upon continuous-time Markov chains (CTMCs). CTMCs are continuous-time processes in which the state $z^{(t)}$ alternates between remaining in the current state and transitioning to another state. The dynamics of the CTMC are governed by a rate matrix $\boldsymbol{R}^{(t)} \in \mathbb{R}^{S \times S}$, where $S$ represents the number of possible states. We denote $R^{(t)}(z^{(t)}, \tilde{z})$ the transition rate from the state $z^{(t)}$ to another state $\tilde{z}$.

Precisely, a CTMC satisfies three differential equations:

$$\text{(forward)} \qquad \partial_t q_{t|s}(\tilde{z} \mid z) = \sum_y q_{t|s}(y \mid z) R^{(t)}(y, \tilde{z}),$$

$$\text{(backward)} \qquad \partial_s q_{t|s}(z \mid \tilde{z}) = \sum_y R^{(s)}(\tilde{z}, y) q_{t|s}(z \mid y),$$

$$\text{(forward, marginals)} \qquad \partial_t q_t(z) = \sum_y q_t(y) R^{(t)}(y, z),$$

where $z, \tilde{z}, y \in \mathcal{Z}$ and $t, s \in [0,1]$. The infinitesimal probability of transition from $z^{(t)}$ to another state $\tilde{z} \in \mathcal{Z}$, for a infinitesimal time step $dt$ between time $t$ and $t + dt$ is given by

$$q_{t+dt|t}(\tilde{z} \mid z^{(t)}) := \begin{cases} R^{(t)}(z^{(t)}, \tilde{z})dt & \text{if } \tilde{z} \neq z^{(t)} \\ 1 + R^{(t)}(z^{(t)}, \tilde{z})dt & \text{if } \tilde{z} = z^{(t)} \end{cases}$$

$$= \delta_{\tilde{z}, z^{(t)}} + R^{(t)}(z^{(t)}, \tilde{z})dt.$$

The rate matrix must satisfy the following conditions:

$$R^{(t)}(z^{(t)}, \tilde{z}) \geq 0, \quad \text{and} \quad R^{(t)}(z^{(t)}, \tilde{z}^{(t)}) = -\sum_{\tilde{z}} R^{(t)}(z^{(t)}, \tilde{z}) < 0.$$

The second condition ensures that $q_{t+dt|t}(\cdot \mid z^{(t)})$ sums to 1. Intuitively, the rate matrix contains instantaneous transition rates, i.e., the number of transitions per unit of time. Therefore, the higher $R^{(t)}(z^{(t)}, \tilde{z})$, the more likely the transition from $z^{(t)}$ to $\tilde{z}$. Since the diagonal coefficients of the rate matrix can be derived from the off-diagonal ones, we will define only the latter in the following.

The CTMC is initialized with $q(z^{(0)}) = p_{\text{data}}(z^{(0)})$. The key challenge in designing the rate matrix is ensuring that the forward process converges to a well-known categorical distribution, that we can later use as prior distribution during the generative process. In the discrete case, such a distribution can be,

e.g., a uniform distribution, a discretized-Gaussian distribution, or an absorbing distribution (Campbell et al., 2022).

The reverse process can also be formulated as a CTMC. Using similar notations than for the forward process, the reverse process is defined through

$$q_{t|t+dt}(z^{(t)} \mid \tilde{z}) = \delta_{z^{(t)}, \tilde{z}} + \hat{R}^{(t)}(\tilde{z}, z^{(t)})dt,$$

where $\hat{\boldsymbol{R}}^{(t)}$ is the rate matrix for the reverse process. Similar to discrete-time diffusion models, this reverse rate can be expressed as (Campbell et al., 2022, Proposition 1):

$$\hat{R}^{(t)}(\tilde{z}, z^{(t)}) = R^{(t)}(z^{(t)}, \tilde{z}) \sum_{z^{(0)} \in \mathcal{Z}} \frac{q_{t|0}(\tilde{z} \mid z^{(0)})}{q_{t|0}(z^{(t)} \mid z^{(0)})} q_{0|t}(z^{(0)} \mid z^{(t)}), \quad \text{for } z \neq \tilde{z}.$$

Since the true reverse process $q_{0|t}(z^{(0)} \mid z^{(t)})$ is intractable, we approximate it using a neural network $p_{0|t}^{\theta}(z^{(0)} \mid z)$ parameterized by $\theta$, yielding the approximate reverse rate:

$$\hat{R}^{t,\theta}(z, \tilde{z}) = R^{(t)}(\tilde{z}, z) \sum_{z^{(0)} \in \mathcal{Z}} \frac{q_{t|0}(\tilde{z} \mid z^{(0)})}{q_{t|0}(z \mid z^{(0)})} p_{0|t}^{\theta}(z^{(0)} \mid z), \quad \text{for } z \neq \tilde{z}.$$

Diffusion models are typically optimized by minimizing the negative ELBO on the negative log-likelihood, $-\log p_0^{\theta}(z^{(0)})$. Campbell et al. (2022, Proposition 2) provides an expression for the ELBO. Although it is not used in this work, we include it for completeness:

$$\mathcal{L}_{CT}(\theta) = T \, \mathbb{E}_{t \sim \mathcal{U}([0,T]), q_t(z), r(z|\tilde{z})} \left( \sum_{z' \neq \tilde{z}} \hat{R}^{t,\theta}(z, z') - \mathcal{Z}^{(t)} \log \left( \hat{R}^{t,\theta}(\tilde{z}, z) \right) \right) + C,$$

where $C$ is a constant independent of $\theta$, $\mathcal{Z}^{(t)} = \sum_{z' \neq z} R^{(t)}(z, z')$, and $r(z \mid \tilde{z}) = 1 - \delta_{\tilde{z}, z} R^{(t)}(z, \tilde{z})/\mathcal{Z}^{(t)}$.

For efficient optimization, it is essential to express $q_t(z) = q_{t|0}(z \mid z^{(0)}) q_0(z^{(0)})$ in closed form. In this context, the transition matrix $\boldsymbol{R}^{(t)}$ must be designed so that $q_{t|0}(z \mid z^{(0)})$ has a closed-form expression. Campbell et al. (2022) established that when $\boldsymbol{R}^{(t)}$ and $\boldsymbol{R}^{(t')}$ commute for any $t$ and $t'$, the transition probability matrix can be written as:

$$\bar{\boldsymbol{Q}}^{(t)} := q_{t|0}(z^{(t)} \mid z^{(0)}) = \exp\left( \int_0^t R^{(s)} ds \right),$$

where $(\bar{\boldsymbol{Q}}^{(t)})_{ij} = q(z^{(t)} = i \mid z^0 = j)$. This condition is met when the rate is written as $\boldsymbol{R}^{(t)} = \beta(t) \boldsymbol{R}_b$, where $\beta$ is a time-dependent scalar and $\boldsymbol{R}_b$ is a constant base rate matrix. In that case, the forward process can be further refined as:

$$(\bar{\boldsymbol{Q}}^{(t)})_{kl} = q_{t|0}(z^{(t)} = k \mid z^{(0)} = l) = \left( \boldsymbol{P} \exp\left[ \boldsymbol{\Lambda} \int_0^t \beta(s) ds \right] \boldsymbol{P}^{-1} \right)_{kl},$$

where $\boldsymbol{R}_b = \boldsymbol{P} \boldsymbol{\Lambda} \boldsymbol{P}^{-1}$ and $\exp$ refers to the element-wise exponential.

Given a one-hot encoded data sample $\boldsymbol{z}^{(0)}$, we can sample a noisy state $\boldsymbol{z}^{(t)}$ by sampling a categorical distribution with probability vector $\boldsymbol{z}^{(0)} \bar{\boldsymbol{Q}}^{(t)}$.

Since most real-world data is multi-dimensional, the above framework needs to be extended to $D$ dimensions. This is done by assuming that each dimension is noised independently so that the forward process factorizes as

$$q_{t+dt|t}(\tilde{\boldsymbol{z}} \mid \boldsymbol{z}) = \prod_{d=1}^{D} q_{t+dt|t}(\tilde{z}_d \mid z_d),$$

where $q_{t+\Delta t|t}(\tilde{z}_d \mid z_d)$ is the unidimensional forward process on the $d$th dimension.

Campbell et al. (2022, Proposition 3) establishes how to write the forward and reverse rates in the $D$-dimensional case:

$$R^{(t)}(\tilde{z}, z) = \sum_{d=1}^{D} \delta_{\tilde{z}^{\backslash d}, z^{\backslash d}} R_d^{(t)}(\tilde{z}_d, z_d),$$

$$\hat{R}^{t,\theta}(z, \tilde{z}) = \sum_{d=1}^{D} \delta_{\tilde{z}^{\backslash d}, z^{\backslash d}} R_d^{(t)}(\tilde{z}_d, z_d) \sum_{z_0} \frac{q_{t|0}(\tilde{z}_d \mid z_d^0)}{q_{t|0}(z_d \mid z_d^0)} p_{0|t}^{\theta}(z_d^0 \mid z),$$

for $z_d \neq \tilde{z}_d$. In brief, assuming that a transition cannot occur in two different dimensions simultaneously, the multi-dimensional rates are equal to the unidimensional rates in the dimension of transition. Importantly, if the dimensions are independent in the forward process, they are not in the reverse process since the whole state is given as input in $p_{0|t}^{\theta}(z_d^0 \mid z)$.

Finally, we need a practical way to simulate the reverse process over finite time intervals for $D$-dimensional data. To that extent, we follow Campbell et al. (2022) and use the $\tau$-leaping algorithm. The first step is to sample $z^{(1)}$ from the prior $p_{\text{ref}}(z^{(1)})$. The sampling procedure is as follows. At each iteration, we keep $z^{(t)}$ and $\hat{R}^{t,\theta}(z, \tilde{z})$ constant and simulate the reverse process for a time interval of length $\tau$. It means that we count all the transitions between $t$ and $t - \tau$ and apply them simultaneously.

The number of transitions in each dimension $z_d^{(t)}$ of the current state $z^{(t)}$, between $z_d^{(t)}$ and $\tilde{z}_d$ is Poisson distributed with mean $\tau \hat{R}^{t,\theta}(z_d^{(t)}, \tilde{z}_d)$. In a state space with no ordinal structure, multiple transitions in one dimension are meaningless, and we reject them. In addition, we experiment using the predictor-corrector scheme. After each predictor step using $\hat{R}^{(t),\theta}(z_d^{(t)}, \tilde{z}_d)$, we can also apply several corrector steps using the expression defined in Campbell et al. (2022), i.e., $\hat{R}^{(t),c} = \hat{R}^{(t),\theta} + R^{(t)}$. The transitions using the corrector rate are counted the same way as for the predictor. This rate admits $q_t(z^{(t)})$ as its stationary distribution, which means that applying the corrector steps brings the distribution of noisy graphs at time $t$ closer to the marginal distribution of the forward process.

## A.2 NOISE SCHEDULE

Here, we provide a proof for Proposition 4, as well as some intuition on the prior distribution.

**Proposition 4.** For a CTMC $(z^{(t)})_{t \in [0,1]}$ with rate matrix $\boldsymbol{R}^{(t)} = \beta(t) \boldsymbol{R}_b$ and $\boldsymbol{R}_b = \mathbb{1}\boldsymbol{m}' - \boldsymbol{I}$, the forward process can be written as

$$\bar{\boldsymbol{Q}}^{(t)} = e^{-\bar{\beta}^t} \boldsymbol{I} + (1 - e^{-\bar{\beta}^t}) \mathbb{1}\boldsymbol{m}',$$

where $(\bar{\boldsymbol{Q}}^{(t)})_{ij} = q(z^{(t)} = i \mid z^{(0)} = j)$ and $\bar{\beta}^t = \int_0^t \beta(s)ds$.

*Proof.* Since $\mathbb{1}\boldsymbol{m}'$ is a rank-one matrix with trace 1, it is diagonalizable and has only one non-zero eigenvalue, equal to $\text{tr}(\mathbb{1}\boldsymbol{m}') = 1$. Therefore,

$$\boldsymbol{R}_b = \mathbb{1}\boldsymbol{m}' - \boldsymbol{I} = \boldsymbol{P}\boldsymbol{D}\boldsymbol{P}^{-1} - \boldsymbol{I} = \boldsymbol{P}(\boldsymbol{D} - \boldsymbol{I}))\boldsymbol{P}^{-1},$$

with $\boldsymbol{D} = \text{diag}(1, 0, \ldots, 0)$. Denoting $\bar{\beta}^t = \int_0^t \beta(s)ds$,

$$\begin{aligned}
\bar{\boldsymbol{Q}}^{(t)} &= \boldsymbol{P}\exp\left(\bar{\beta}^t(\boldsymbol{D} - \boldsymbol{I})\right)\boldsymbol{P}^{-1} \\
&= \boldsymbol{P}\left(\boldsymbol{D} - e^{-\bar{\beta}^t}\boldsymbol{I} - e^{-\bar{\beta}^t}\boldsymbol{D}\right)\boldsymbol{P}^{-1} \\
&= e^{-\bar{\beta}^t}\boldsymbol{I} + (1 - e^{-\bar{\beta}^t})\mathbb{1}\boldsymbol{m}'.
\end{aligned}$$

$\square$

We now wish to elaborate on the link between Proposition 4 and the choice of the prior distribution. Recall our noise schedule,

$$\beta(t) = \alpha\frac{\pi}{2}\sin\left(\frac{\pi}{2}t\right) \quad \text{and} \quad \int_0^t \beta(s)ds = \alpha\left(1 - \cos\left(\frac{\pi}{2}t\right)\right).$$

When $t = 1$, it holds that $e^{-\bar{\beta}^t} = e^{-\alpha}$, and therefore $\bar{\boldsymbol{Q}}^{(1)} = e^{-\alpha}\boldsymbol{I} + (1 - e^{-\alpha})\mathbb{1}\boldsymbol{m}'$. When $\alpha$ is large enough, then $e^{-\alpha} \approx 0$ and $\bar{\boldsymbol{Q}}^{(1)} \approx \mathbb{1}\boldsymbol{m}'$. Denoting $(\bar{\boldsymbol{Q}}^{(1)})_j$ the $j$-th row of $\bar{\boldsymbol{Q}}^{(1)}$, it holds that $(\bar{\boldsymbol{Q}}^{(1)})_j = q(z^{(1)} \mid z^{(0)} = j) \approx \boldsymbol{m}$. In other terms, whatever the value $z^{(0)}$, $z^{(1)}$ is sampled from the same categorical distribution with probability vector $\boldsymbol{m}$. Therefore,

$$
\begin{aligned}
q_1(z^{(1)}) &= \sum_{j \in \mathcal{Z}} q_{1|0}(z^{(1)} \mid z^{(0)} = j) q_0(z^{(0)} = j) \\
&\approx \sum_{j \in \mathcal{Z}} m q_0(z^{(0)} = j) \\
&\approx m = p_{\text{ref}}(z^{(1)})
\end{aligned}
$$

In the $D$-dimensional case, since the forward process factorizes, we get

$$
q_{1|0}(\boldsymbol{z}^{(1)} \mid \boldsymbol{z}^{(0)}) = \prod_{d=0}^{D} m = p_{\text{ref}}(\boldsymbol{z}^{(1)}),
$$

where $d \in [\![0, D]\!]$ denotes $d$-th dimension of $\boldsymbol{z} \in \mathcal{Z}^D$.

### A.3 RRWP PROPERTIES

In the following, we formally study the encodings of the RRWP encoding.

**Notations** A *graph* $G$ is a pair $(V(G), E(G))$ with *finite* sets of *vertices* or *nodes* $V(G)$ and *edges* $E(G) \subseteq \{\{u, v\} \subseteq V(G) \mid u \neq v\}$. If not otherwise stated, we set $n := |V(G)|$, and the graph is of *order* $n$. For ease of notation, we denote the edge $\{u, v\} \in E(G)$ by $(u, v)$ or $(v, u)$. An $n$-order attributed graph is a pair $\boldsymbol{G} = (G, \boldsymbol{X}, \mathsf{E})$, where $G = (V(G), E(G))$ is a graph and $\boldsymbol{X} \in \{0, 1\}^{n \times a}$, for $a > 0$, is a *node feature matrix* and $\mathsf{E} \in \{0, 1\}^{n \times n \times b}$, for $b > 0$, is an *edge feature tensor*. Here, we set $V(G) := [\![1, n]\!]$. The *neighborhood* of $v \in V(G)$ is denoted by $N(v) := \{u \in V(G) \mid (v, u) \in E(G)\}$.

In our experiments, we leverage the *relative random-walk probabilites* (RRWP) encoding, introduced in Ma et al. (2023). Denoting $\boldsymbol{A}$ the adjacency matrix of a graph $G$, and $\boldsymbol{D}$ the diagonal degree matrix, and $\boldsymbol{M} = \boldsymbol{D}^{-1}\boldsymbol{A}$ the degree-normalized adjacency matrix, for each pair of nodes $(i, j)$, the RRWP encoding computes

$$
P_{ij}^K := \left[ I_{ij}, M_{ij}, M_{ij}^2, \ldots, M_{ij}^{K-1} \right], \tag{5}
$$

where $K$ refers to the maximum length of the random walks. The entry $P_{ii}^K$ corresponds to the RWSE encoding of node $i$; therefore, we leverage them as node encodings. This encoding alone is sufficient to train our graph diffusion model and attain state-of-the-art results.

In the following, we show that RWPP encoding can (approximately) determine if two nodes lie in the same connected components and approximate the size of the largest connected component.

**Proposition 5.** For $n \in \mathbb{N}$, let $\mathcal{G}_n$ denote the set of $n$-order graphs and for a graph $G \in \mathcal{G}_n$ let $V(G) := [\![1, n]\!]$. Assume that the graph $G$ has $c$ connected components and let $\boldsymbol{C} \in \{0, 1\}^{n \times c}$ be a matrix such the $i$th row $\boldsymbol{C}_{i\cdot}$ is a one-hot encoding indicating which of the $c$ connected components the vertex $i$ belongs to. Then, for any $\varepsilon > 0$, there exists a feed-forward neural network $\mathsf{FNN} \colon \mathbb{R}^{n \times n} \to [0, 1]^{n \times c}$ such that

$$
\|\mathsf{FNN}(\boldsymbol{M}^{n-1}) - \boldsymbol{C}\|_F \leq \varepsilon.
$$

*Proof.* Let $\boldsymbol{R} := \boldsymbol{M}^{n-1}$. First, since the graphs have $n$ vertices, the longest path in the graphs has length $n - 1$. Hence, two vertices $v, w \in V(G)$, with $v \neq w$ are in the same connected component if, and only, if $R_{vw} \neq 0$. Hence, applying a sign activation function to $\boldsymbol{R}$ pointwisely, we get a matrix over $\{0, 1\}$ with the same property. Further, by adding $\boldsymbol{D}_n \in \{0, 1\}^{n \times n}$, an $n \times n$ diagonal matrix with ones on the main diagonal, to this matrix, this property also holds for the case of $v = w$. In addition, there exists a permutation matrix $\boldsymbol{P}_n$ such that applying it to the above matrix results in a block-diagonal matrix $\boldsymbol{B} \in \{0, 1\}^{n \times n}$ such that $\boldsymbol{B}_{v\cdot} = \boldsymbol{B}_{w\cdot}$, for $v, w \in V(G)$, if, and only, if the vertices $v, w$ are in the same connected component. Since $n$ is finite, the number of such $\boldsymbol{B}$ matrices is finite and hence compact. Hence, we can find a continuous function mapping each possible row of $\boldsymbol{B}_{v\cdot}$, for $v \in V(G)$, to the corresponding one-hot encoding of the connected component. Since all functions after applying

the sign function are continuous, we can approximate the above composition of functions via a two-layer feed-forward neural network leveraging the universal approximation theorem (Cybenko, 1992; Leshno et al., 1993). □

Similarly, we can also approximate the size of the largest component in a given graph.

**Proposition 6.** For $n \in \mathbb{N}$, let $\mathcal{G}_n$ denote the set of $n$-order graphs and for $G \in \mathcal{G}_n$ let $V(G) \coloneqq [\![1, n]\!]$. Assume that $S$ is the number of vertices in the largest connected component of the graph $G$. Then, for any $\varepsilon > 0$, there exists a feed-forward neural network $\mathsf{FNN} \colon \mathbb{R}^{n \times n} \to [1, n]$,

$$|\mathsf{FNN}(\boldsymbol{M}^{n-1}) - S| \leq \varepsilon.$$

*Proof.* By the proof of Proposition 5, we a get a block-diagonal matrix $\boldsymbol{B} \in \{0, 1\}^{n \times n}$, such that $B_{uv} = 1$ if, and only, if $u, v$ are in the same connected components. Hence, by column-wise summation, we get the number of vertices in each connected component. Hence, there is an $n \times 1$ matrix over $\{0, 1\}$, extracting the largest entry. Since all of the above functions are continuous, we can approximate the above composition of functions via a two-layer feed-forward neural network leveraging the universal approximation theorem (Cybenko, 1992; Leshno et al., 1993). □

Moreover, we can show RRWP encodings can (approximately) count the number $p$-cycles, for $p < 5$, in which a node is contained. A $p$-cycle is a cycle on $p$ vertices.

**Proposition 7.** For $n \in \mathbb{N}$, let $\mathcal{G}_n$ denote the set of $n$-order graphs and for $G \in \mathcal{G}_n$ let $V(G) \coloneqq [\![1, n]\!]$. Assume that $\boldsymbol{c} \in \mathbb{N}^n$ contains the number of $p$-cycles a node is contained in for all vertices in $G$, for $p \in \{3, 4\}$. Then, for any $\varepsilon > 0$, there exists a feed-forward neural network $\mathsf{FNN} \colon \mathbb{R}^{n \times n} \to \mathbb{R}^n$,

$$\|\mathsf{FNN}(\boldsymbol{P}^{n-1}) - \boldsymbol{c}\|_2 \leq \varepsilon.$$

*Proof.* For $p \in \{3, 4\}$, Vignac et al. (2022, Appendix B.2) provide simple linear-algebraic equations for the number of $p$-cycles each vertex of a given graph is contained based on powers of the adjacency matrix, which can be expressed as compositions of linear mappings, i.e., continuous functions. Observe that we can extract these matrices from $\boldsymbol{P}^{n-1}$. Further, note that the domain of these mappings is compact. Hence, we can approximate this composition of functions via a two-layer feed-forward neural network leveraging the universal approximation theorem (Cybenko, 1992; Leshno et al., 1993). □

However, we can also show that RRWP encodings cannot detect if a node is contained in a large cycle of a given graph. We say that an encoding, e.g., RRWP, counts the number of $p$-cycles for $p \geq 2$ if there do not exist two graphs, one containing at least one $p$-cycle while the other does not, while the RRWP encodings of the two graphs are equivalent.

**Proposition 8.** For $p \geq 8$, the RRWP encoding does not count the number of $p$-cycles.

*Proof.* First, by Rattan and Seppelt (2023), the RRWP encoding does not distinguish more pairs of non-isomorphic graphs than the so-called $(1, 1)$-dimensional Weisfeiler–Leman algorithm. Secondly, the latter algorithm is strictly weaker than the 3-dimensional Weisfeiler–Leman algorithm in distinguishing non-isomorphic graphs (Rattan and Seppelt, 2023, Theorem 1.4). However, by Fürer (2017, Theorem 4), the 3-dimensional Weisfeiler–Leman algorithm cannot count 8-cycles. □

Hence, the above proposition implies the following results.

**Corollary 9.** For $p \geq 8$ and $K \geq 0$, there exists a graph $G$ containing a $p$-cycle $C$, and two vertex pairs $(r, s), (v, w) \in V(G)^2$ such that $(r, s)$ is contained in $C$ while $(v, w)$ is not and $P_{vw}^K = P_{rs}^K$.

### A.4 EQUIVARIANCE PROPERTIES

In this section, we prove that our model is equivariant (Proposition 10) and that our loss is permutation-invariant (Proposition 11), relying on Vignac et al. (2022, Lemma 3.1 and 3.2). We also prove exchangeability with Proposition 12.

Let us start by defining the notation for a *graph permutation*. Denote $\pi$ a permutation, $\pi$ acts on the attributed graph $\boldsymbol{G} = (G, \boldsymbol{X}, \mathbf{E})$ as,

- $\pi G = (V(\pi G), E(\pi G))$ where $V(\pi G) = \{\pi(1), \ldots, \pi(n)\}$ and $E(\pi G) = \{(\pi(i), \pi(j)) \mid (v_i, v_j) \in E(G)\}$,

- $\pi \boldsymbol{X}$ the matrix obtained by permutating the rows of $\boldsymbol{X}$ according to $\pi$, i.e. $(\pi \boldsymbol{X})_i = \boldsymbol{x}_{\pi^{-1}(i)}$,

- Similarly, $\pi \boldsymbol{E}$ is the tensor obtained by the permutation of the components $e_{ij}$ of E according to $\pi$, i.e. $(\pi \boldsymbol{E})_{ij} = \boldsymbol{e}_{\pi^{-1}(i)\pi^{-1}(j)}$.

**Proposition 10** (Equivariance). DIGRESS' graph transformer using RWSE as node encodings and RRWP as edge encodings is permutation equivariant.

*Proof.* We recall the sufficient three conditions stated in Vignac et al. (2022) for ensuring permutation-equivariance of the DIGRESS architecture, namely,

- their set of structural and spectral features is equivariant.

- All the blocks of their graph transformer architecture are permutation equivariant.

- The layer normalization is equivariant.

Replacing the first condition with the permutation-equivariant nature of the RRWP-based node and edge encodings completes the proof. $\square$

We now derive a more thorough proof of the permutation invariance of the loss compared to Vignac et al. (2022, Lemma 3.2), relying on the permutation-equivariant nature of both the forward process and the denoising neural network.

**Proposition 11** (Permutation invariance of the loss). The cross-entropy loss defined in Equation 3 is invariant to the permutation of the input graph $G^{(0)}$.

*Proof.* Given a graph $\boldsymbol{G} = (G, \boldsymbol{X}, \mathbf{E})$, we denote by $\hat{\boldsymbol{G}} = (\hat{G}, \hat{\boldsymbol{X}}, \hat{\mathbf{E}})$ the predicted clean graph by the neural network and $\pi \boldsymbol{G} = (\pi G, \pi \boldsymbol{X}, \pi \boldsymbol{E})$ a permutation of this graph, for arbitrary permutation $\pi$. Let us now establish that the loss function is permutation-invariant. We recall the loss function for a permutation $\pi$ of the clean data sample $G^{(0)}$ is

$$\mathcal{L}_{\text{CE}} := \mathbb{E}_{t \sim [0,1], p_{\text{data}}(\pi G^{(0)}), q(\pi G^{(t)} | \pi G^{(0)})} \left[ -\sum_i^n \log p_{0|t}^\theta(x_{\pi(i)}^{(0)} \mid \pi G^{(t)}) - \lambda \sum_{i<j}^n \log p_{0|t}^\theta(e_{\pi(i)\pi(j)}^{(0)} \mid \pi G^{(t)}) \right].$$

Because dimensions are noised independently, the true data distribution $p_{\text{data}}(\pi G^{(0)}) = p_{\text{data}}(G^{(0)})$ is permutation-invariant, and the forward process is permutation-equivariant. Thus, we can write,

$$\mathcal{L}_{\text{CE}} := \mathbb{E}_{t \sim [0,1], p_{\text{data}}(G^{(0)}), q(G^{(t)} | G^{(0)})} \left[ -\sum_i^n \log p_{0|t}^\theta(x_{\pi(i)}^{(0)} \mid \pi G^{(t)}) - \lambda \sum_{i<j}^n \log p_{0|t}^\theta(e_{\pi(i)\pi(j)}^{(0)} \mid \pi G^{(t)}) \right].$$

Using Proposition 10, we also have that $p_{0|t}^\theta(x_i^{(0)} \mid G^{(t)}) = p_{0|t}^\theta(x_{\pi(i)}^{(0)} \mid \pi G^{(t)})$ and $p_{0|t}^\theta(e_{ij}^{(0)} \mid G^{(t)}) = p_{0|t}^\theta(e_{\pi(i)\pi(j)}^{(0)} \mid \pi G^{(t)})$, which concludes the proof. $\square$

Proposition 11 shows that, whatever permutation of the original graph we consider, the loss function remains the same, and so do the gradients. Hence, we do not have to consider all the permutations of the same graph during the optimization process.

**Proposition 12** (Exchangeability). COMETH yields exchangeable distributions.

*Proof.* To establish the exchangeability, we require two conditions, a permutation-invariant prior distribution and an equivariant reverse process.

- Since nodes and edges are sampled i.i.d from the same distribution, our prior distribution is permutation-invariant, i.e., each permutation of the same random graph has the same probability of being sampled. Hence $p_{\text{ref}}(\pi G^{(T)}) = p_{\text{ref}}(G^{(T)})$.

- It is straightforward to see that our reverse rate is permutation-equivariant regarding the joint permutations of $G^{(t)}$ and $G^{(0)}$. We illustrate this using the node reverse rate,

$$R_X^t(\tilde{x}_i, x_i^{(t)}) \sum_{x_0} \frac{q_{t|0}(\tilde{x}_i \mid x_i^{(0)})}{q_{t|0}(x_i^{(t)} \mid x_i^{(0)})} p_{0|t}^\theta(x_i^{(0)} \mid \boldsymbol{G}^{(t)}).$$

The forward rate, as well as the forward process, is permutation-equivariant regarding the joint any permutation on $G^{(t)}$ and $G^{(0)}$, and the neural network is permutation-equivariant. Similarly, we can reason regarding the edge reverse rate. Therefore, the overall reverse rate is permutation-equivariant. Since we sample independently across dimensions, the $\tau$-leaping procedure is also permutation-equivariant.

$\square$

## A.5 CLASSIFIER-FREE GUIDANCE

In the conditional generation setting, one wants to generate samples satisfying a specific property $\boldsymbol{y}$, to which we refer as *the conditioner*. For example, in text-to-image diffusion models, the conditioner consists of a textual description specifying the image the model is intended to generate. The most straightforward way to perform conditional generation for diffusion models is to inject the conditioner into the network—therefore modeling $p^\theta(z^{(t-1)} \mid z^{(t)}, \boldsymbol{y})$—hoping that the model will take it into account. However, the network might ignore $\boldsymbol{y}$, and several efficient approaches to conditional generation for diffusion models were consequently developed.

The approach leveraged by Vignac et al. (2022) to perform conditional generation is classifier-guidance. It relies on a trained unconditional diffusion model and a regressor, or classifier, depending on the conditioner, trained to predict the conditioner given noisy inputs. As mentioned in Ho and Salimans (2021), it has the disadvantage of complicating the training pipeline, as a pre-trained classifier cannot be used during inference.

To avoid training a classifier to guide the sampling process, *classifier-free guidance* has been proposed in Ho and Salimans (2021) and then adapted for discrete data in Tang et al. (2022). A classifier-free conditional diffusion model jointly trains a conditional and unconditional model through *conditional dropout*. That is, the conditioner is randomly dropped with probability $p_{\text{uncond}}$ during training, in which the conditioner is set to a null vector. However, Tang et al. (2022) showed that learning the null conditioner jointly with the model's parameters is more efficient.

At the sampling stage, the next state is sampled through

$$\log p^\theta(z^{(t-1)} \mid z^{(t)}, \boldsymbol{y}) = \log p^\theta(z^{(t-1)} \mid z^{(t)}, \emptyset) + (s+1)(\log p^\theta(z^{(t-1)} \mid z^{(t)}, \boldsymbol{y}) - \log p^\theta(z^{(t-1)} \mid z^{(t)}, \emptyset)), \tag{6}$$

where $s$ is the *guidance strength*. We refer to Tang et al. (2022) for deriving the above expression for the sampling process.

Let us now explain how we apply classifier-free guidance in our setting. Denoting $\hat{R}^{t,\theta}(\boldsymbol{G}, \tilde{\boldsymbol{G}} \mid \boldsymbol{y})$, the conditional reverse rate can be written as

$$\hat{R}^{t,\theta}(\boldsymbol{G}, \tilde{\boldsymbol{G}} \mid \boldsymbol{y}) = \sum_i \delta_{\boldsymbol{G}^{\backslash x_i}, \tilde{\boldsymbol{G}}^{\backslash x_i}} \hat{R}_X^{t,\theta}(x_i^{(t)}, \tilde{x} \mid \boldsymbol{y}) + \sum_{i<j} \delta_{\boldsymbol{G}^{\backslash e_{ij}}, \tilde{\boldsymbol{G}}^{\backslash e_{ij}}} \hat{R}_E^{t,\theta}(e_{ij}^{(t)}, \tilde{e}_{ij} \mid \boldsymbol{y}),$$

and

$$\hat{R}_X^{t,\theta}(x_i^{(t)}, \tilde{x}) = R_X^t(\tilde{x}_i, x_i^{(t)}) \sum_{x_0} \frac{q_{t|0}(\tilde{x}_i \mid x_i^{(0)})}{q_{t|0}(x_i^{(t)} \mid x_i^{(0)})} p_{0|t}^\theta(x_i^{(0)} \mid \boldsymbol{G}^{(t)}, \boldsymbol{y}), \text{ for } x_i^{(t)} \neq \tilde{x}_i,$$

and similarly for edges. At the sampling stage, we first compute the unconditional probability distribution $p_{0|t}^\theta(x_i^{(0)} \mid \boldsymbol{G}^{(t)}, \emptyset)$, where $\emptyset$ denotes the learned null vector, then the conditional distribution $p_{0|t}^\theta(x_i^{(0)} \mid \boldsymbol{G}^{(t)}, \boldsymbol{y})$. These two distributions are combined in the log-probability space in the following way,

$$\log p^\theta(z^{(t-1)} \mid z^{(t)}, \boldsymbol{y}) = \log p^\theta(z^{(t-1)} \mid z^{(t)}, \emptyset) + (s+1)(\log p^\theta(z^{(t-1)} \mid z^{(t)}, \boldsymbol{y}) - \log p^\theta(z^{(t-1)} \mid z^{(t)}, \emptyset)). \tag{7}$$

Finally, the distribution in Equation (7) is exponentiated and plugged into the reverse rate.

## B  IMPLEMENTATION DETAILS

Here, we provide some implementation details.

### B.1  REVERSE PROCESS : TAU-LEAPING AND PREDICTOR-CORRECTOR

At the sampling stage, we use the $\tau$-leaping algorithm to generate new samples, as proposed by Campbell et al. (2022). We first sample the graph size and then sample the noisy graph $G^{(t)}$ from the prior distribution. At each iteration, we keep $G^{(t)}$ and $\hat{R}^{t,\theta}(G, \tilde{G})$ constant and simulate the reverse process for a time interval of length $\tau$. In practice, we count all the transitions between $t$ and $t - \tau$ and apply them simultaneously. The number of transitions between $x_i^{(t)}$ and $\tilde{x}_i$ (respectively $e_{ij}^{(t)}$ and $\tilde{e}_{ij}$) is Poisson distributed with mean $\tau R_X^{(t),\theta}(x_i^{(t)}, \tilde{x})$ (respectively $\tau \hat{R}_E^{(t),\theta}(e_{ij}^{(t)}, \tilde{e}_{ij})$). Since our state space has no ordinal structure, multiple transitions in one dimension are meaningless, and we reject them. In addition, we experiment using the predictor-corrector scheme. After each predictor step using $\hat{R}^{(t),\theta}(G, \tilde{G})$, we can also apply several corrector steps using the expression defined in Campbell et al. (2022), i.e., $\hat{R}^{(t),c} = \hat{R}^{(t),\theta} + R^{(t)}$. This rate admits $q_t(\mathbf{G}^{(t)})$ as its stationary distribution, which means that applying the corrector steps brings the distribution of noisy graphs at time $t$ closer to the marginal distribution of the forward process. As $\tau$ is fixed only during the sampling stage, its value can be adjusted to balance sample quality and efficiency, i.e., the number of model evaluations. We perform such an ablation study in Appendix D.

### B.2  CONDITIONAL GENERATION

If an unconditional generation is essential to designing an efficient diffusion model, conditioning the generation on some high-level property is critical in numerous real-world applications Corso et al. (2023); Lee and Min (2022). In addition, Vignac et al. (2022) used *classifier guidance*, which relies on a trained unconditional model guided by a regressor on the target property. However, to our knowledge, classifier guidance has yet to be adapted to continuous-time discrete-state diffusion models. We, therefore, leverage another approach to conditional diffusion models, *classifier-free guidance* (Tang et al., 2022), for which we provide a detailed description in Appendix A.5.

### B.3  NOISE SCHEDULE

We plot our noise schedule against the constant noise schedule used for categorical data in Campbell et al. (2022) in Figure 2. Following Proposition 1, we plot $\bar{\alpha}_t = e^{-\bar{\beta}_t}$ on the $y$-axis, quantifying the information level of the original data sample retained at time $t$. Similarly to Nichol and Dhariwal (2021), we can see that the constant noise schedule converges towards zero faster than the cosine schedule, hence degrading the data faster. In our experiments, we perform a hyperparameter search to select the best rate constant for each dataset, with $\alpha \in \{4, 5, 6\}$. Following Campbell et al. (2022); we set a minimum time to $t_{\min} = 0.01T$ because the reverse rates are ill-conditioned close to $t = 0$.

### B.4  ALGORITHMS

We provide the pseudo-code for the training and sampling from COMETH in Figure 3. Similar to Campbell et al. (2022), we apply a last neural network pass at $t = t_{\min}$ and set the node and edge types to the types with the highest predicted probability. We omit the corrector steps in the sampling algorithm for conciseness. They are exactly the same as the predictor $\tau$-leaping steps, using the corrector rate $\hat{R}^{(t),c} = \hat{R}^{(t),\theta} + R^{(t)}$, and applied after time update, i.e. at $t - \tau$. Since those steps are sampled from different CTMC with rate $\hat{R}^{(t),c}$, we have control over $\tau$ when applying corrector steps. We provide additional details on the choice of this hyperparameter, denoted as $\tau_c$, in Appendix C.

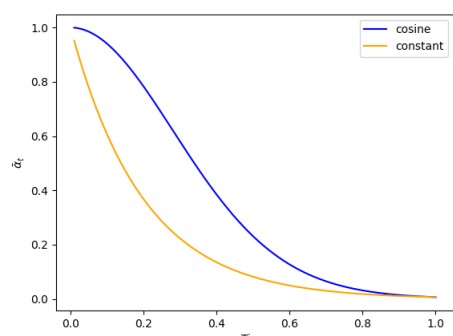

Figure 2: Comparison between our cosine noise schedule and the constant noise schedule proposed by Campbell et al. (2022). Both schedules are plotted using a rate constant $\alpha = 5$.

### B.5 GRAPH TRANSFORMER

See Figure 4 for an overview of the used graph transformer, building on Vignac et al. (2022). We use the RRWP encoding, defined in Equation 5, for synthetic graph generation. For molecule generation datasets, we additionally compute several molecular features used in Vignac et al. (2022), namely the valency and charge for each node and the molecular weight.

## C EXPERIMENTAL DETAILS

Here, we outline the details of your experimental study.

### C.1 SYNTHETIC GRAPH GENERATION

We evaluate our method on two datasets from the SPECTRE benchmark (Martinkus et al., 2022), with 200 graphs each. PLANAR contains planar graphs of 64 nodes, and SBM contains graphs drawn from a stochastic block model with up to 187 nodes. We use the same split as the original paper, which uses 128 graphs for training, 40 for training, and the rest as a validation set. Similar to Jo et al. (2024), we apply random permutations to the graphs at each training epoch.

We report five metrics from the SPECTRE benchmark, which include four **MMD** metrics between the test set and the generated set and the **VUN** metric on the generated graphs. The MMD metrics measure the **Maximum Mean Discrepancy** between statistics from the test and the generated set, namely the **Degree (Degree)** distribution, the **Clustering coefficient (Cluster)** distribution, the count of orbit information regarding subgraphs of size four **Orbit (Orb.)** and the **eigenvalues (Spectrum)** of the graph Laplacian. The **Valid, Unique, and Novel (VUN)** metric measures the percentage of valid, unique, and non-isomorphic graphs to any graph in the training set.

On PLANAR, we report results using $\tau = 0.002$, i.e. using 500 $\tau$-leaping. We also evaluate our model using 10 corrector steps after each predictor step when $t < 0.1T$, with $\tau = 0.002$, for a total of 1000 $\tau$-leaping steps. We found our best results using $\tau_c = 0.7$.

On SBM, we report results using $\tau = 0.001$, i.e., using 1 000 $\tau$-leaping steps.

### C.2 SMALL MOLECULE GENERATION : QM9

We evaluate our model on QM9 (Wu et al. (2018)) to assess the ability of our model to model attributed graph distributiond. The molecules are kekulized using the RDKit library and hydrogen atoms are removed, following the standard preprocessing pipeline for this dataset. Edges can have three types, namely simple bonds, double bonds, and triple bonds, as well as one additional type for the absence of edges. The atom types are listed in Table 6.

---

**Algorithm 1:** Training

---

**Input:** A graph $G = (\boldsymbol{X}, \boldsymbol{E})$
Sample $t \sim \mathcal{U}([0, 1])$
Sample $G^t \sim \boldsymbol{X}\bar{\boldsymbol{Q}}_X^t \times \boldsymbol{E}\bar{\boldsymbol{Q}}_E^t$     ▷ Sample sparse noisy graph
Predict $p_{0|t}^\theta(\boldsymbol{G} \mid \boldsymbol{G}^{(t)})$   ▷ Predict clean graph using neural network
$\mathcal{L}_{CE} \leftarrow -\sum_i^n \log p_{0|t}^\theta(x_i^{(0)} \mid \boldsymbol{G}^{(t)}) - \lambda \sum_{i<j}^n \log p_{0|t}^\theta(e_{ij}^{(0)} \mid \boldsymbol{G}^{(t)})$
Update $\theta$ using $\mathcal{L}_{CE}$

---

**Algorithm 2:** $\tau$-leaping sampling of Cometh

---

Sample $n$ from the training data distribution
Sample $G^{(T)} \sim \prod_i m_X \prod_{ij} m_E$   ▷ Sample random graph from prior
 distribution
**while** $t > 0.01$ **do**
 **for** $i = 1$ **to** $n$ **do**
  **for** $\tilde{x}$ *in* $\mathcal{X}$ **do**
   $\hat{R}_X^{t,\theta}(x_i^{(t)}, \tilde{x}) = R_X^t(\tilde{x}_i, x_i^{(t)}) \sum_{x_0} \frac{q_{t|0}(\tilde{x}_i|x_i^{(0)})}{q_{t|0}(x_i^{(t)}|x_i^{(0)})} p_{0|t}^\theta(x_i^{(0)} \mid G^{(t)})$, for $x_i^{(t)} \neq \tilde{x}_i$
   Sample $j_{x_i^{(t)}, \tilde{x}} \sim \mathcal{P}(\tau\hat{R}_X^{t,\theta}(x_i^{(t)}, \tilde{x}))$  ▷ Count transitions on node i
  **end**
 **end**
 **for** $i, j = 1$ **to** $n$, $i < j$ **do**
  **for** $\tilde{e}$ *in* $\mathcal{E}$ **do**
   $\hat{R}_E^{t,\theta}(e_{ij}^{(t)}, \tilde{e}_{ij}) = R_E^t(\tilde{e}_{ij}, e_{ij}^{(t)}) \sum_{e_0} \frac{q_{t|0}(\tilde{e}_{ij}|e_{ij}^{(0)})}{q_{t|0}(e_{ij}^{(t)}|e_{ij}^{(0)})} p_{0|t}^\theta(e_{ij}^{(0)} \mid G^{(t)})$, for $e_{ij}^{(t)} \neq \tilde{e}_{ij}$
   Sample $j_{e_{ij}^{(t)}, \tilde{e}_{ij}} \sim \mathcal{P}(\tau\hat{R}_E^{t,\theta}(e_{ij}^{(t)}, \tilde{e}_{ij}))$ ▷ Count transitions on edge ij
  **end**
 **end**
 **for** $i = 1$ **to** $n$ **do**
  **for** $\tilde{x}$ *in* $\mathcal{X}$ **do**
   **if** $j_{x_i^{(t)}, \tilde{x}} = 1$ *and* $\sum_{\tilde{x}} j_{x_i^{(t)}, \tilde{x}} = 1$ **then**
    $x_i^{(t-\tau)} = \tilde{x}$    ▷ Apply unique transition or discard
   **end**
  **end**
 **end**
 **for** $i, j = 1$ **to** $n$, $i < j$ **do**
  **for** $\tilde{e}$ *in* $\mathcal{E}$ **do**
   **if** $j_{e_{ij}^{(t)}, \tilde{e}} = 1$ *and* $\sum_{\tilde{e}} j_{e_{ij}^{(t)}, \tilde{e}} = 1$ **then**
    $e_{ij}^{(t-\tau)} = \tilde{e}$    ▷ Apply unique transition or discard
   **end**
  **end**
 **end**
 $t \leftarrow t - \tau$
**end**
$G^0 \leftarrow \prod_i \text{argmax}\, p_{0|t}^\theta(x_i^{(0)} \mid G^{(t)}) \prod_{ij} \text{argmax}\, p_{0|t}^\theta(e_{ij}^{(0)} \mid G^{(t)})$    ▷ Last pass
**return** $G^0$

---

Figure 3: Training and Sampling algorithms of COMETH

We use the same split as Vignac et al. (2022), i.e., 100k molecules for training, 13k for testing, and the rest (20 885 molecules) as a validation set. We choose this split over the one proposed in Jo et al. (2022) because it leaves a validation set to evaluate the ELBO and select the best checkpoints to minimize this quantity. In consequence, our training dataset contains roughly 20k molecules, which is less than what most graph generation works use.

At the sampling stage, we generate 10k molecules. We evaluate four metrics. The **Validity** is evaluated by sanitizing the molecules and converting them to SMILES string using the RDKit library. The largest molecular fragment is selected as a sample if it is disconnected. We then evaluate the **Uniqueness** among valid molecules. As stated in Vignac et al. (2022), evaluating novelty on QM9 bears little sense since this dataset consists of an enumeration of all stable molecules containing the atom above types with size nine or smaller. We also evaluate the **Fréchet ChemNet Distance (FCD)**, which embeds the generated set and the test set using the ChemNet neural network and compares the resulting distributions using the Wasserstein-2 distance (Preuer et al. (2018)). Finally, we evaluate the *Neighborhood Subgraph*

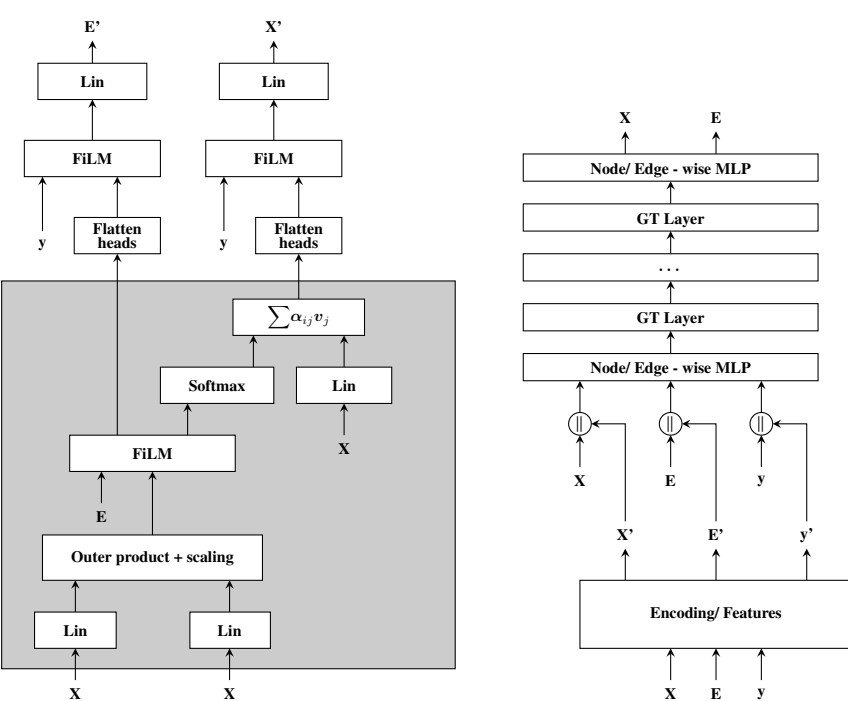

Figure 4: Overview of DIGRESS graph transformer.

*Pairwise Distance Kernel* (NSPDK) between the test set and the generated, which measures the structural similarities between those two distributions.

## C.3  MOLECULE GENERATION ON LARGE DATASETS

We further evaluate COMETH on two large molecule generation benchmarks, MOSES (Polykovskiy et al., 2020) and GuacaMol (Brown et al., 2019). The molecules are processed the same way as for QM9 and present the same edge types. The atom types for both datasets are listed in Table 6. The filtration procedure for the GuacaMol consists of converting the SMILES into graphs and retrieving the original SMILES. The molecules for which this conversion is not possible are discarded. We use the code of Vignac et al. (2022) to perform this procedure. We use the standard split provided for each dataset.

Both datasets are accompanied by their own benchmarking libraries. For GuacaMol, we use the distribution learning benchmark. During the sampling stage, we generate 25k molecules for MOSES and 18k for GuacaMol, which is sufficient for both datasets, as they evaluate metrics based on 10k molecules sampled from the generated SMILES provided.

We then elaborate on the metrics for each dataset. For both datasets, we report **Validity**, defined in the same manner as for QM9, the percentage of **Valid and Unique (Val. & Uni.)** samples, and the percentage of **Valid, Unique, and Novel (VUN)** samples. We prefer the latter two metrics over Uniqueness and Novelty alone, as they provide a better assessment of a model's performance compared to separately reporting all three metrics (Validity, Uniqueness, and Novelty). The MOSES benchmark also computes metrics by comparing the generated set to a scaffold test set, from which we report the **Fréchet ChemNet Distance (FCD)**, the **Similarity to the nearest neighbor (SNN)**, which computes the average Tanimoto similarity between the molecular fingerprints of a generated set and the fingerprints of the molecules of a reference set, and **Scaffold similarity (Scaf)**, which compares the frequencies of the Bemis-Murcko scaffolds in the generated set and a reference set. Finally, we report the **Filters** metrics, which indicate the percentage of generated molecules successfully passing the filters applied when constructing the dataset. The GuacaMol benchmark computes two *scores*, the **Fréchet ChemNet Distance (FCD)** score and the **KL divergences (KL)** between the distributions of a set of physicochemical descriptors in the training set and the generated set.

Table 6: Details of the molecular datasets. The number of molecules for GuacaMol is computed after filtration.

| Dataset | Number of molecules | Size | Atom types |
|---|---|---|---|
| QM9 | 133 885 | 1 to 9 | C, N, O, F |
| MOSES | 1 936 962 | 8 to 27 | C, N, S, O, F, Cl, Br |
| GuacaMol | 1 398 223 | 2 to 88 | C, N, O, F, B, Br, Cl, I, P, S, Se, Si |

We report results using $\tau = 0.002$, i.e., 500 denoising steps on both datasets. The experiments using the predictor-corrector were performed using $\tau = 0.002$ and 10 corrector steps for a total of 500 denoising steps. For both datasets, we used $\tau_c = 1.5$.

### C.4 CONDITIONAL GENERATION

We perform conditional generation experiments on QM9, targeting two properties, the **dipole moment** $\mu$ and the **highest occupied molecular orbital energy (HOMO)**. They are well suited for conditional generation evaluation because they can be estimated using the Psi4 library (Smith et al., 2020). We trained models sweeping over $p_{\text{uncond}} \in \{0.1, 0.2\}$, and explore different values for $s$ in $[\![1, 6]\!]$ during sampling. We obtained our best results using $p_{\text{uncond}} = 0.1$ and $s = 1$.

During inference, we evaluated our method in the same setting as Vignac et al. (2022). We sampled 100 molecules from the test set, extracted their dipole moment and HOMO values, and generated 10 molecules targeting those properties. We estimated the HOMO energy and the dipole moment of the generated molecules, and we report the **Mean Absolute Error (MAE)** between the estimated properties and the corresponding targets.

To efficiently incorporate the conditioner $\boldsymbol{y}$, we implemented a couple of ideas proposed in Ninniri et al. (2023). Instead of using $\boldsymbol{y}$ solely as a global feature, we incorporated it as an additional feature for each node and edge. Additionally, we trained a two-layer neural network to predict the size of the molecule given the target properties rather than sampling it from the empirical dataset distribution. Our empirical observations indicate that this approach enhances performance. As of the time of writing, no official implementation has been released for Ninniri et al. (2023), rendering it impossible to reproduce their results. Additionally, since they do not report validity in their experiments on QM9, we choose not to include their results as a baseline to avoid unfair comparisons.

### C.5 COMPUTE RESSOURCES

Experiments on QM9, PLANAR, and SBM were carried out using a single V100 or A10 GPU at the training and sampling stage. The training time on QM9 is 6 hours, while the training time on SBM and Planar is approximately 2 days and a half.

We trained models on MOSES or Guacamol using two A100 GPUs. To sample from these models, we used a single A100 GPU. The training time on MOSES is approximately two days, while training on GuacaMol required 4 days.

## D ADDITIONAL EXPERIMENTS

### D.1 ABLATION ON THE NUMBER OF STEPS

To demonstrate why the continuous-time approach allows trade sampling quality and efficiency, we perform an ablation study on $\tau$. Results are presented in Tables 7 to 9. We report the number of model evaluations equal to $1/\tau$ instead of $\tau$ for readability.

Overall, we observe that the model achieves decent performance across all datasets with just 50 steps. Increasing the number of model evaluations to 500-700 enhances performance to a state-of-the-art level. Beyond this point, performance saturates, and models using 1000 steps do not necessarily outperform those with fewer evaluations, as seen with SBM and PLANAR.

Table 7: **Ablation study on the number of steps for synthetic graphs**. We report the mean of 5 runs, as well as 95% confidence intervals. The best results are highlighted in bold.

| Number of steps | Degree ↓ | Cluster ↓ | Orbit ↓ | Spectrum ↓ | VUN [%] ↑ |
|---|---|---|---|---|---|
| | | *Planar graphs* | | | |
| 10 | $103.0_{\pm6.8}$ | $11.0_{\pm0.1}$ | $305.4_{\pm10.6}$ | $5.3_{\pm0.2}$ | $0.0_{\pm0.0}$ |
| 50 | $3.6_{\pm1.3}$ | $3.3_{\pm0.2}$ | $12.4_{\pm4.5}$ | $1.3_{\pm0.2}$ | $41.5_{\pm4.51}$ |
| 100 | $3.1_{\pm1.2}$ | $2.3_{\pm0.3}$ | $5.2_{\pm2.5}$ | $1.4_{\pm0.2}$ | $76.0_{\pm2.23}$ |
| 300 | $3.4_{\pm1.1}$ | $1.7_{\pm0.5}$ | $6.2_{\pm3.9}$ | $1.3_{\pm0.1}$ | $86.5_{\pm3.82}$ |
| 500 | $2.1_{\pm1.3}$ | $\mathbf{1.5}_{\pm0.3}$ | $3.1_{\pm0.3}$ | $1.2_{\pm0.2}$ | $92.5_{\pm3.67}$ |
| 700 | $2.4_{\pm1.2}$ | $\mathbf{1.5}_{\pm0.2}$ | $\mathbf{2.2}_{\pm1.2}$ | $\mathbf{1.1}_{\pm0.2}$ | $\mathbf{94.0}_{\pm2.23}$ |
| 1000 | $\mathbf{1.9}_{\pm1.0}$ | $1.9_{\pm0.2}$ | $2.7_{\pm1.7}$ | $1.5_{\pm0.2}$ | $89.5_{\pm3.51}$ |
| | | *Stochastic block model* | | | |
| 10 | $166.6_{\pm16.1}$ | $1.8_{\pm0.1}$ | $3.3_{\pm0.2}$ | $2.2_{\pm0.2}$ | $14.0_{\pm4.72}$ |
| 50 | $2.6_{\pm0.6}$ | $\mathbf{1.5}_{\pm0.0}$ | $1.9_{\pm0.2}$ | $0.9_{\pm0.1}$ | $62.0_{\pm5.44}$ |
| 100 | $\mathbf{1.4}_{\pm0.7}$ | $\mathbf{1.5}_{\pm0.0}$ | $1.7_{\pm0.2}$ | $0.9_{\pm0.1}$ | $70.5_{\pm5.26}$ |
| 300 | $2.4_{\pm0.6}$ | $\mathbf{1.5}_{\pm0.0}$ | $1.8_{\pm0.3}$ | $\mathbf{0.8}_{\pm0.0}$ | $65.5_{\pm5.94}$ |
| 500 | $2.4_{\pm1.1}$ | $\mathbf{1.5}_{\pm0.0}$ | $1.7_{\pm0.2}$ | $\mathbf{0.8}_{\pm0.1}$ | $\mathbf{77.0}_{\pm5.26}$ |
| 700 | $2.6_{\pm0.9}$ | $\mathbf{1.5}_{\pm0.0}$ | $\mathbf{1.6}_{\pm0.1}$ | $0.9_{\pm0.1}$ | $69.0_{\pm4.06}$ |
| 1000 | $1.8_{\pm0.7}$ | $\mathbf{1.5}_{\pm0.0}$ | $1.7_{\pm0.4}$ | $\mathbf{0.8}_{\pm0.1}$ | $67.5_{\pm3.10}$ |

Table 8: **Ablation study on the number of steps for QM9.** We report the mean of 5 runs, as well as 95% confidence intervals.

| Number of steps | Validity ↑ | Uniqueness ↑ | Valid & Unique ↑ | FCD ↓ | NSPDK ↓ |
|---|---|---|---|---|---|
| 10 | $88.69_{\pm0.36}$ | $98.57_{\pm0.13}$ | $87.42_{\pm0.46}$ | $0.84_{\pm0.02}$ | $0.001_{\pm0.0}$ |
| 50 | $99.07_{\pm0.05}$ | $96.78_{\pm0.16}$ | $95.88_{\pm0.21}$ | $0.25_{\pm0.01}$ | $0.000_{\pm0.0}$ |
| 100 | $99.42_{\pm0.06}$ | $96.81_{\pm0.06}$ | $96.24_{\pm0.05}$ | $0.26_{\pm0.01}$ | $0.000_{\pm0.0}$ |
| 300 | $99.53_{\pm0.02}$ | $96.57_{\pm0.12}$ | $96.12_{\pm0.11}$ | $0.25_{\pm0.01}$ | $0.000_{\pm0.0}$ |
| 500 | $99.57_{\pm0.07}$ | $96.76_{\pm0.17}$ | $96.34_{\pm0.2}$ | $0.25_{\pm0.01}$ | $0.000_{\pm0.0}$ |
| 700 | $99.53_{\pm0.05}$ | $96.65_{\pm0.15}$ | $96.2_{\pm0.15}$ | $0.25_{\pm0.01}$ | $0.000_{\pm0.0}$ |
| 1000 | $99.57_{\pm0.07}$ | $96.79_{\pm0.08}$ | $96.37_{\pm0.14}$ | $0.25_{\pm0.01}$ | $0.000_{\pm0.0}$ |

Table 9: **Ablation study on the number of steps for MOSES** We report the mean of five runs, as well as 95% confidence intervals. The best results are highlighted in bold.

| Number of steps | Val. ↑ | Val. & Uni. ↑ | VUN ↑ | Filters ↑ | FCD ↓ | SNN ↑ | Scaf ↑ |
|---|---|---|---|---|---|---|---|
| 10 | $26.1_{\pm0.2}$ | $26.1_{\pm0.2}$ | $26.0_{\pm0.2}$ | $59.9_{\pm0.6}$ | $7.88_{\pm0.13}$ | $0.36_{\pm0.0}$ | $8.9_{\pm1.1}$ |
| 50 | $82.9_{\pm0.3}$ | $82.9_{\pm0.3}$ | $80.5_{\pm0.3}$ | $94.6_{\pm0.1}$ | $1.54_{\pm0.01}$ | $0.49_{\pm0.0}$ | $18.4_{\pm1.0}$ |
| 100 | $85.8_{\pm0.2}$ | $85.7_{\pm0.1}$ | $82.9_{\pm0.2}$ | $96.5_{\pm0.1}$ | $\mathbf{1.43}_{\pm0.01}$ | $0.5_{\pm0.0}$ | $17.2_{\pm0.6}$ |
| 300 | $86.9_{\pm0.2}$ | $86.9_{\pm0.2}$ | $83.8_{\pm0.2}$ | $97.1_{\pm0.1}$ | $1.44_{\pm0.02}$ | $\mathbf{0.51}_{\pm.0}$ | $\mathbf{17.8}_{\pm1.0}$ |
| 500 | $87.0_{\pm0.2}$ | $86.9_{\pm0.2}$ | $83.8_{\pm0.2}$ | $\mathbf{97.2}_{\pm0.1}$ | $1.44_{\pm0.02}$ | $\mathbf{0.51}_{\pm0.0}$ | $15.9_{\pm0.8}$ |
| 700 | $\mathbf{87.2}_{\pm0.2}$ | $87.1_{\pm0.2}$ | $83.9_{\pm0.2}$ | $\mathbf{97.2}_{\pm0.1}$ | $\mathbf{1.43}_{\pm0.02}$ | $\mathbf{0.51}_{\pm0.0}$ | $15.9_{\pm0.4}$ |
| 1000 | $\mathbf{87.2}_{\pm0.2}$ | $\mathbf{87.2}_{\pm0.2}$ | $\mathbf{84.0}_{\pm0.2}$ | $\mathbf{97.2}_{\pm0.1}$ | $1.44_{\pm0.01}$ | $\mathbf{0.51}_{\pm0.0}$ | $17.3_{\pm0.9}$ |

## D.2 ABLATION ON THE NOISE MODEL

To emphasize the impact of using marginal transitions instead of uniform transitions, we trained models on PLANAR and SBM with the uniform noise model (see table 10). While the uniform model performs competitively with marginal transitions on SBM, the VUN score remains significantly higher with marginal transitions. On PLANAR, marginal transitions demonstrate a substantially superior performance compared to uniform transitions.

## D.3 ABLATION ON THE POSITIONAL ENCODING

We conducted an ablation study on the positional encoding to compare the benefits of the RRWP encoding against the feature set used in Digress (see table 11). While both approaches achieve comparable results across most distribution metrics, RRWP significantly outperforms DiGress' set of features in terms of VUN.

Table 10: **Ablation study on the noise model for synthetic graphs**. We report the mean of 5 runs, as well as 95% confidence intervals.

| Noise Model | Degree ↓ | Cluster ↓ | Orbit ↓ | Spectrum ↓ | VUN [%] ↑ |
|---|---|---|---|---|---|
| *Planar graphs* | | | | | |
| Marginal | $2.1_{\pm1.3}$ | $1.5_{\pm0.3}$ | $3.1_{\pm3.0}$ | $1.2_{\pm0.2}$ | $92.5_{\pm3.67}$ |
| Uniform | $14.3_{\pm4.2}$ | $3.8_{\pm0.6}$ | $14.2_{\pm6.1}$ | $1.7_{\pm0.2}$ | $32.5_{\pm4.4}$ |
| *Stochastic block model* | | | | | |
| Marginal | $2.4_{\pm1.1}$ | $1.5_{\pm0.0}$ | $1.7_{\pm0.2}$ | $0.8_{\pm0.1}$ | $77.0_{\pm5.26}$ |
| Uniform | $1.6_{\pm0.3}$ | $1.5_{\pm0.0}$ | $1.8_{\pm0.3}$ | $0.9_{\pm0.1}$ | $63.5_{\pm6.6}$ |

Table 11: **Ablation study on the positional encoding for synthetic graphs**. We report the mean of 5 runs, as well as 95% confidence intervals.

| Noise Model | Degree ↓ | Cluster ↓ | Orbit ↓ | Spectrum ↓ | VUN [%] ↑ |
|---|---|---|---|---|---|
| *Planar graphs* | | | | | |
| RRWP | $2.1_{\pm1.3}$ | $1.5_{\pm0.3}$ | $3.1_{\pm3.0}$ | $1.2_{\pm0.2}$ | $92.5_{\pm3.67}$ |
| DiGress' features | $2.2_{\pm1.1}$ | $2.2_{\pm0.3}$ | $18.0_{\pm7.4}$ | $1.3_{\pm0.2}$ | $67.5_{\pm3.7}$ |
| *Stochastic block model* | | | | | |
| Marginal | $2.4_{\pm1.1}$ | $1.5_{\pm0.0}$ | $1.7_{\pm0.2}$ | $0.8_{\pm0.1}$ | $77.0_{\pm5.26}$ |
| DiGress' features | $2.3_{\pm1.2}$ | $1.5_{\pm0.0}$ | $1.3_{\pm0.2}$ | $0.9_{\pm0.2}$ | $64.5_{\pm6.41}$ |

## D.4 ABLATION ON THE LOSS FUNCTION

To better justify our choice to use the cross-entropy as our loss function instead of the ELBO, we provide the results of our experiments on QM9 using the ELBO as our loss function. Figure 5 presents the validation performance of both approaches regarding Validity, over 512 samples. While the cross-entropy loss allows to quickly reach a near-perfect Validity, the model trained using the ELBO saturates below 80%. The significant performance gap on a simple dataset like QM9 underscores the inefficiency of using the ELBO as the loss function for Cometh.

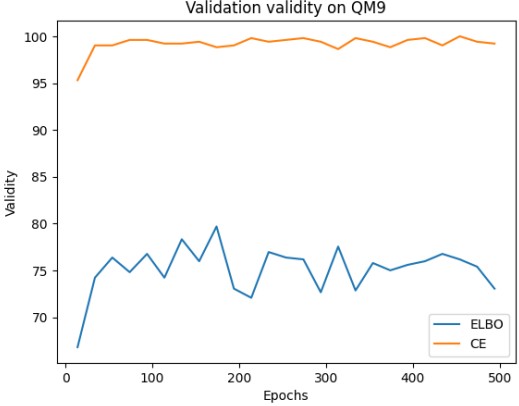

Figure 5: Validation results on QM9 using the cross-entropy loss and the ELBO loss.

## D.5 ABLATION ON THE NOISE SCHEDULE

We also performed a simple ablation study on QM9 to compare the performance of our cosine noise schedule against the exponential noise schedule $\beta(t) = \alpha\gamma^t \log\gamma$, using $\alpha = 0.8$ and $\gamma = 2$. Overall, our cosine schedule performs better on every metric, except the Uniqueness.

Table 12: **Ablation on the noise schedule on QM9.** We report the mean of five runs, as well as 95% confidence intervals. Best results are highlighted in bold.

| Model | Validity ↑ | Uniqueness ↑ | Valid & Unique ↑ | FCD ↓ | NSPDK ↓ |
|---|---|---|---|---|---|
| COSINE | $99.57_{\pm 0.07}$ | $96.76_{\pm 0.17}$ | $\mathbf{96.34}_{\pm 0.2}$ | $\mathbf{0.25}_{\pm 0.01}$ | $\mathbf{0.000}_{\pm 0.00}$ |
| EXP | $98.28_{\pm 0.15}$ | $\mathbf{97.0}_{\pm 0.13}$ | $95.34_{\pm 0.15}$ | $0.31_{\pm 0.01}$ | $\mathbf{0.001}_{\pm 0.00}$ |

## E  LIMITATIONS

Although our model advances the state-of-the-art across all considered benchmarks, it still faces quadratic complexity, a common issue in graph diffusion models. This problem could be alleviated by adapting methods like EDGE (Chen et al., 2023) used to scale DIGRESS for large graph generation. Additionally, our approach does not support the generation of continuous features and is restricted to categorical attributes. To generate continuous features, it should be combined with a continuous-state diffusion model, resulting in an approach similar to Vignac et al. (2023).

## F  SAMPLES

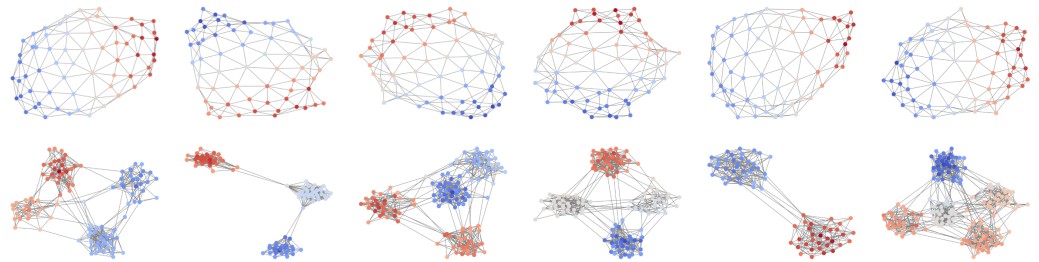

Figure 6: Samples from COMETH on PLANAR (top) and SBM (bottom)

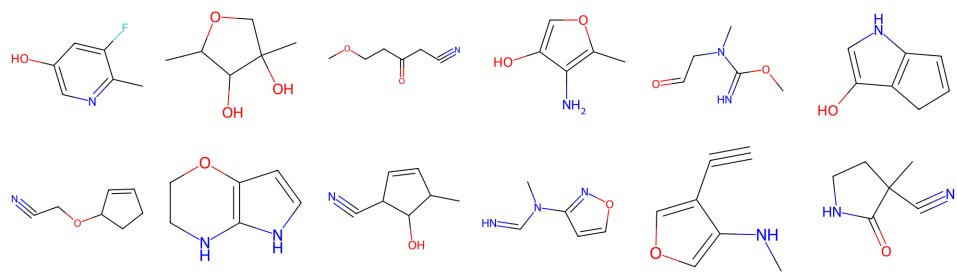

Figure 7: Samples from COMETH on QM9.

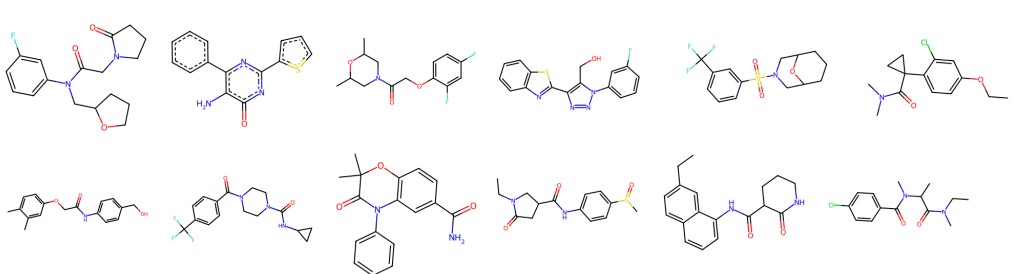

Figure 8: Samples from COMETH on MOSES.

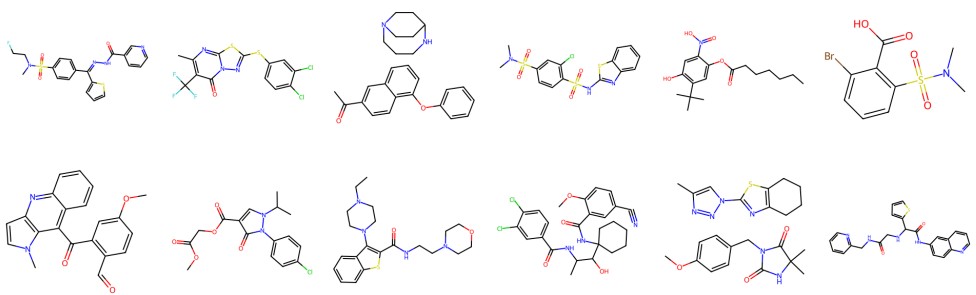

Figure 9: Samples from COMETH on GuacaMol. The samples on this dataset exhibit some failure cases, such as disconnected molecules or 3-cycles of carbon atoms.

