# OpenReview forum: "Cometh: A continuous-time discrete-state graph diffusion model"
_ICLR.cc/2025/Conference — Submitted to ICLR 2025_

### Official Review · Reviewer_QUdV · 2024-10-27

**Soundness:** 3
**Presentation:** 3
**Contribution:** 2
**Rating:** 5
**Confidence:** 4

**Summary:**

The paper introduces a graph generation model named COMETH, which combines the advantages of discrete state denoising diffusion models with continuous-time dynamics. COMETH integrates graph data into a continuous-time diffusion framework, achieving a better balance between sampling efficiency and quality. Experimental evidence shows that this continuous-time integration significantly improves performance on various molecular and non-molecular benchmark datasets compared to related graph generation models.

**Strengths:**

1. This paper proposes a new continuous-time discrete state diffusion model framework, integrating discrete state diffusion models into a continuous-time framework to leverage the efficiency enhancement techniques of continuous diffusion models during training and sampling.
2. This model demonstrates strong performance across multiple benchmark datasets, surpassing state-of-the-art models in both molecular and non-molecular graph synthesis.

**Weaknesses:**

1. The proposed framework mainly combines existing techniques, such as continuous-time Markov chains (CTMC) with marginal transitions and random-walk encoding.
2. There is a lack of comparison with continuous diffusion models on large-scale datasets like MOSES and GuacaMol.
3. Although Appendix D demonstrates the performance of COMETH with fewer number of steps, there is a lack of comparison with the training and sampling times of related works.

**Questions:**

Mentioned in Weaknesses.

---

> ### Author Response · Authors · 2024-11-21
>
> We thank you for reviewing our work, and provide answers to your concerns below.
>
> **The proposed framework mainly combines existing techniques, such as continuous-time Markov chains (CTMC) with marginal transitions and random-walk encoding.** While our work builds largely on Campbell et al. (2022), Vignac et al. (2022), and Ma et al. (2023), we emphasize that the extensions were far from straightforward. For example, the marginal transition could not be directly extended from Campbell et al. (2022), requiring us to derive Proposition 1 to enable the use of arbitrary target distributions and to design a new noise schedule. Regarding the random-walk encoding, this is its first application in a generative setting, as well as the first instance of using a graph positional encoding with provable expressivity in graph diffusion models.
>
> **There is a lack of comparison with continuous diffusion models on large-scale datasets like MOSES and GuacaMol.** There is not a single continuous-state graph diffusion model; GruM is one example of such an approach. To the best of our knowledge, no such models have been evaluated on MOSES and GuacaMol, and performing such computationally expensive evaluations falls outside the scope of our paper.
> The MOSES and GuacaMol benchmarks include their own set of baselines. Our objective in conducting these experiments was to highlight two key points:
> - Performance Comparison: Cometh outperforms its discrete-time counterpart.
> - Benchmark Integration: Cometh further bridges the gap between graph diffusion models and the leading baselines on these datasets.
>
> **Although Appendix D demonstrates COMETH's performance with fewer steps, the training and sampling times of related works are not compared.** We acknowledge that we could add the time requirements of our models and some time comparisons with DiGress. We will add this in a future revised version.
>
> If you are satisfied with our responses, please consider adjusting your score. We are happy to answer any remaining questions. Thank you.

---

> > ### Comment · Reviewer_QUdV · 2024-11-25
> > **Thank you for the response**
> >
> > Thank you for the response. I remain my original rating.

---

### Official Review · Reviewer_hvvy · 2024-10-28

**Soundness:** 3
**Presentation:** 3
**Contribution:** 1
**Rating:** 3
**Confidence:** 4

**Summary:**

The authors propose a continuous-time discrete-state diffusion model for graphs. As an additional contribution, the authors replace the graph-specific features often used to augment GNN-based denoisers (e.g. in Vignac et al. (2022) [1]) by a more-principled random-walk encoding.

**Strengths:**

- **Clarity of presentation**: the paper has a clear goal (introducing continuous time discrete-state graph diffusion to enable additional sampling methods) and suggests a suitable approach to achieve it (adapting the continuous-time framework proposed by Campbell et al. (2022) [2])
- **Empirical improvement**: COMETH seems to be competitive with existing baselines, achieving better results on at least a couple of metrics in all experiments.

**Weaknesses:**

The main weakness of this work is its incremental nature. Below I analyze the contributions of the work and explain my assessment.

**TL;DR** Since the continuous time framework for discrete data has already been proposed, a good direction for the present work is to provide a thorough evaluation of the design choices relevant to adapting this framework to graph diffusion. This can include for example both theoretical and empirical evaluations of the choice of loss function, noise schedule, and sampling procedures, to only name a few. A theoretical analysis of why continuous time can benefit graph diffusion in particular could also be beneficial here.

- **Limited technical novelty in adapting the continuous-time framework**: The paper applies Campbell et al. (2022)'s [2] continuous-time discrete-state framework to graph diffusion by treating nodes and edges as independent components. This adaptation follows naturally from Campbell's CTMC for handling independent dimensions and was already established for graphs in discrete time by Vignac et al. (2022) [1]. The paper does not introduce fundamental changes to either framework, making this contribution largely incremental. Perhaps a thorough empirical study demonstrating how the continuous-time process is a critical choice for graph-diffusion could make this contribution more relevant.
- **Insufficient justification for loss function choice**: The authors replace Campbell's ELBO loss with cross-entropy, citing only that "preliminary experiments using this loss led to poor empirical results" (page 5). No theoretical analysis is provided for why ELBO performs poorly or what guarantees might be lost by this substitution. While following Vignac et al. (2022)'s [1] choice of cross-entropy, this seems more like an engineering decision than a principled modification of the continuous-time framework. A theoretical (and empirical) analysis of the effects of each loss could be beneficial here.
- **Discussion of the noise schedule is misleading**: The authors claim that Campbell et al. (2022) [2] "used a constant noise schedule for categorical data" which they replace with a cosine schedule to ensure more gradual decay of data (Figure 2, Appendix B.1.). However, this misrepresents Campbell's work - their main text does not prescribe constant schedules, and they successfully use an exponential schedule β(t) = ab^t log(b) in most experiments. The only mention of a constant schedule appears in Appendix H.3 for one music generation experiment, noted as "sufficient for this dataset." By overstating Campbell's commitment to constant schedules, the authors inflate their switch to a cosine schedule as addressing a fundamental limitation, when it is simply a design choice made without theoretical or empirical justification over the exponential alternative. I recommend rephrasing the discussion of the noise schedule in the main text and the appendix to rectify this situation.
- **Proposition 1 is an adaptation of DiGress[1]'s forward process formulation** Proposition 1 adapts DiGress[1]'s forward process to continuous time through a direct substitution of rate matrices, without leveraging any continuous-time properties. The proof relies solely on matrix algebra, making this change incremental rather than a fundamental theoretical contribution.

- **The properties of the random walk encoding follow directly from Ma et al. (2023) [3]** the properties in question are the size of the largest component, whether two nodes lie in the same component, and counting the number of p-cycles in which a node is contained.
    - While the first property extends Ma et al. [3]'s results from 'RWPP encodes distances' to 'RWPP captures structural properties', the result itself still follows from an argument regarding distances. A similar remark was made by Ma et al. [3] briefly in page 4: "[...] for K = n, we recover all shortest path distances (with disconnected nodes getting a distance of n, which is higher than the maximum distance n − 1 between connected nodes)."
    - The second property is a direct consequence of the first: we can count the size of the largest component if we know the connectivity of all the nodes.
    - The estimation of p-cycles uses DiGress [1]'s inference of p-cycles from powers of the adjacency matrix of the graph, which can be directly deduced from the random walk encoding. The result regarding the inability to identify p-cycles larger than 8 comes from an interesting combination of previous results, but constitutes a minor contribution in itself.
    - Even if these new properties are worth stating explicitly beyond the results of Ma et al. (2023) [3], I do not think this constitutes an important enough contribution for the present work, not to mention that it is not in fact its main focus. If the authors would like to emphasize this contribution, they can change the text of the paper to bring more focus to it, perhaps also comparing to other encodings in the process.
- **Absence of code/implementation**: it would be nice if the authors could share an implementation of their method + checkpoints for the model to replicate their results.
- **Improving presentation**: I believe including additional figures can make the main text richer and improve its flow. For instance, the authors could add figures showing the effect of denoising steps on the sample quality of COMETH, the denoising progression on example graphs for COMETH vs DiGress (to highlight the benefits of continuous time processes), etc.

**Questions:**

**clarifications**
- Can you explain why cross-entropy is a better loss when diffusing graphs compared to other discrete data? Is anything lost by foregoing ELBO in favor of CE? Sharing results comparing both approaches would be helpful as well.

- Can you expand on your noise schedule choice as it relates to graph diffusion in continuous time in particular? Why do you think the cosine schedule fits better here compared to other schedules? Empirical evidence could be useful.

- Could you elaborate on why the random walk encoding is a better choice than other encodings (e.g. Laplacian)?

- Can you discuss why you think the random walk encoding is a sufficient replacement for the additional features used by DiGress? Namely, DiGress uses features beyond the graph structure (e.g. chemical properties of molecules), which might still be beneficial to the denoiser even with a continuous time process.

**suggestions**
- The authors should include campbell 2022 [2] in the related work section since their framework is heavily adapted in COMETH.
- I found figure 1 a bit confusing. Perhaps the 'sample G^(t - tau)' can be directly between 'compute R' and 'predict G(0)'?

**nitpicks**
- I recommend adding numbers to all equations.
- The first sentence in 'Present work' (first page) => graph is duplicated
- The first line in section **2. Background** has 'discrete-state' repeated twice.
- The numbers in table 3 and 4 are missing bolding
- The second part in the equation given in A.1 after "The rate matrix must satisfy the following conditions:" the second z(t) should be \tilde{z}
- The last sentence in 4.3 is ambiguous: 'This may be due to the fact that we train on a subset of the original dataset, whereas the LSTM is trained directly on SMILES.': is  the difference with LSTM the data modality or the training subsets?

---

> ### Author Response · Authors · 2024-11-21
>
> We would like to thank for taking the time to write such a detailed review and for your insightful feebacks. We
>
> **Limited technical novelty in adapting the continuous-time framework** Although we agree that we build on tools from Campbell et al. 2022, we would like to highlight that the extension of marginal transitions does not trivially derive from their formulation. Indeed, the link between the chosen rate matrix and the prior distribution was not explicitly established. In that regard, the result established in Proposition 1 allow for using any target distribution and open the design space of possible noise schedules. Hence, would like to argue that our contribution is not just a simple extension of Campbell et al. 2022.
>
> Additionally, we believe our experimental results prove the superiority of Cometh over its discrete time counterpart, DiGress, in terms of performance.
>
> **Insufficient justification for loss function choice** We acknowledge the fact that using direct model supervision with the cross-entropy requires more theoretical justifications. Campbell et al. 2022 provide some insights concerning direct model supervision in the appendix of their paper, by proving that the ideal denoiser of their framework minimizes the cross-entropy loss. They also prove that in *discrete time*, minimizing the cross-entropy loss amounts to minimizing an upper bound on the ELBO, i.e., a looser bound on the NNL than the ELBO. Though they don’t provide the same result in continuous time, this is another theoretical hint that indicates why CE can be a good candidate for loss function.
>
> Also, in the revised version of the manuscript, we will provide results using the ELBO as a loss function to justify our claim better.
>
> **Discussion of the noise schedule is misleading** We apologize for this misunderstanding and will rewrite this paragraph to better reflect the true nature of our contribution. Additionally, we will perform an ablation study to compare our cosine schedule against the exponential schedule
>
> **Proposition 1 is an adaptation of DiGress[1]'s forward process formulation** We are afraid this is a misunderstanding. Proposition 1 is derived from the expression of the forward process formulated in Campbell et al, 2022. It is however true that the resulting expression resembles the forward process of discrete-time models.
>
> Also, the proof is indeed rather simple, but we believe that this result, which was not explicitly derived in Campbell et al. 2022, is fundamental to easily understanding the interplay between the choice of rate matrix, the noise schedule, and the behavior of the forward process. The formulation of the forward process in Campbell et al. 2022 did not allow for an easy derivation of the expression of p(z^T), and the behavior of the forward process was hardly interpretable. Conversely, Proposition 1 provides a clear interpretability of the role of each component in the forward process and allows us to easily design each of them separately.
>
> **The properties of the random walk encoding follow directly from Ma et al. (2023)** We recognize that the results are a relatively straightforward extension of Ma et al. (2023). However, we find them noteworthy, especially given the newly identified limitation of RRWP. Taken together, these findings illustrate why RRWP can effectively replace most of DiGress’s structural features. Moreover, positional encoding is a well-explored topic in graph learning, and our work serves to align graph diffusion models with discriminative models in this context.
>
> **Absence of code/implementation** We will provide the code and a checkpoint so that you reviewers can reproduce our results.
>
> **Can you expand on your noise schedule choice related to graph diffusion in continuous time in particular?** Our choice of noise schedule was simply made to match older heuristics of discrete time diffusion models. We also demonstrate with Proposition 1 how any noise schedule converging towards 1 when t is close to 1 can be used. We therefore open the design space of noise schedules for discrete diffusion models and showcasts this using the cosine schedule.
>
> **Could you elaborate on why the random walk encoding is a better choice than other encodings (e.g. Laplacian)?** The main advantage of RRWP is that it allows it to encompass most auxiliary features used in DiGress – namely the cycles and connected components features – but using only one, easily computed encoding.
>
> Some Laplacian features are also used in DiGress.Although RRWP does not directly encompass those features, the RWSE component within RRWP, as explained in line 311, serves as an effective alternative to them. In fact, RWSE is a commonly used node-level encoding, which has performed on par with Laplacian encodings in various graph learning settings (​e.g. Rampasek et al., 2022).

---

> > ### Author Response · Authors · 2024-11-21
> >
> > **DiGress uses features beyond the graph structure (e.g. chemical properties of molecules)** This is an omission from our side. DiGress indeed uses valency and molecular weight as auxiliary domain-specific features for molecular experiments, and we use them as well. We will add the information in a revised version of the manuscript. However, note that no ablation study has ever been performed to study the impact of those two molecular features. Anyway, concerning the valency, it is likely that it can be learned through the second iteration of RRWP, which is the adjacency matrix.
> >
> > If you are satisfied with our responses, please consider adjusting your score. Thank you.

---

> ### Comment · Reviewer_hvvy · 2024-11-25
>
> Thank you for taking the time to write a detailed rebuttal. I am afraid I cannot change my score before the changes and results requested are provided.
>
> In the meantime, I have some follow-ups to the authors' response:
>
> - **Proposition 1 being derived from Campbell et al. (2022)**: Thank you for clarifying this point. While the derivation proposed in the proposition allows for interpretability, I believe the proposition itself still constitutes a minor technical contribution within the framework of continuous time diffusion.
>
> - **RRWP and replacing DiGress's features**: the ability to obtain multiple features from a single encoding is indeed interesting to note, but as pointed out in my initial review, reaching this observation does not require novel technical contributions on top of the work of Ma et al. 2023. The limitations of RRWP similarly build on established results in the literature. Therefore, I maintain that this is not a substantial  contribution of COMETH.

---

> > ### Author Response · Authors · 2024-11-25
> >
> > > RRWP and replacing DiGress's features: the ability to obtain multiple features from a single encoding is indeed interesting to note, but as pointed out in my initial review, reaching this observation does not require novel technical contributions on top of the work of Ma et al. 2023. The limitations of RRWP similarly build on established results in the literature. Therefore, I maintain that this is not a substantial contribution of COMETH.
> >
> > While we agree that our results studying the limitations build on established results, we believe that it is important to state these results formally to show that RRWP has severe limitations.
> >
> > > Proposition 1 being derived from Campbell et al. (2022): Thank you for clarifying this point. While the derivation proposed in the proposition allows for interpretability, I believe the proposition itself still constitutes a minor technical contribution within the framework of continuous time diffusion.
> >
> > We will make this more clear in the final version of the paper.
> >
> > Please let us know if you have further questions. We will submit the revised version soon.

---

> > > ### Comment · Reviewer_hvvy · 2024-11-26
> > >
> > > Thank for providing an updated version of your manuscript. Unfortunately, my concerns regarding the limited novelty of this work persist. Therefore, I would like to maintain my score.

---

### Official Review · Reviewer_5bR2 · 2024-10-31

**Soundness:** 3
**Presentation:** 3
**Contribution:** 2
**Rating:** 3
**Confidence:** 5

**Summary:**

This paper proposes a generative model for graph data using a discrete state space within a continuous time framework. It extends the previously proposed marginal transition noisy scheme, used for generating graphs, to a continuous-time setting. Additionally, the paper adopts a random-walk-based feature, which appears to enhance performance to some extent. Empirically, this approach achieves good results on synthetic and one molecular benchmark.

**Strengths:**

1. This paper have achieved state-of-the-art results on synthetic and one of the molecular benchmarks.
2. It makes notable machine learning contributions by adapting continuous-time discrete diffusion from [1] to develop a rate matrix, resulting in a noise model of marginal transition. The paper proposes using RRWP features to effectively model the graph, and appears to enhance performance to some extent.

**Weaknesses:**

1. From a machine learning perspective, the contribution of the proposed extension for marginal transition seems minor, and its impact on improving graph generation remains unclear.
2. While the RRWP feature seems to enhance generation performance, the lack of an ablation study makes it difficult to determine the extent to which each component contributes and which factors are responsible for the improvements.
3. There are inconsistencies in the abbreviations "RWPP" and "RRWP" on lines #314 and #847. Additionally, the last minus sign in the proof of Proposition 1 on line #82 appears to be incorrect.
4. The structure of the paper could be improved by making "Related Work" an independent section.

**Questions:**

1. On line #84, why is the discrete-time sampling scheme referred to as ancestral sampling, while continuous-time sampling is not?
2. How does the continuous-time Markov chain (CTMC) version of discrete-state marginal transition diffusion aid in graph generation?
3. On line #178, why is the difference between nodes and edges considered a challenge when the treatment appears to be the same throughout the paper?
4. The model performs well on QM9 but not as well on the other two molecular datasets. What might be the reason for this discrepancy?

[1] Andrew Campbell, Joe Benton, Valentin De Bortoli, Tom Rainforth, George Deligiannidis, Arnaud Doucet. "A Continuous Time Framework for Discrete Denoising
Models"

---

> ### Author Response · Authors · 2024-11-21
>
> We thank you for you constructive feedback. Below, we address your concern and answer your questions about our work.
>
> **the contribution of the proposed extension for marginal transition seems minor, and its impact on improving graph generation remains unclear.** Using marginal transitions with a marginal prior distribution is standard in graph diffusion models (Vignac et al. 2022, Qin et al. 2023), as we elaborate in l.202. The marginal prior is optimal within our chosen distribution space, being the closest product-form distribution to the data distribution. This avoids issues like the uniform prior, which would result in overly dense graphs (e.g., a density of 0.5 for Planar compared to its actual 0.9). While we rely on tools from Campbell et al. 2022, the extension of marginal transitions is non-trivial, as the link between the rate matrix and the prior was not explicitly established, making our contribution more than a straightforward extension.
>
> To better highlight the importance of extending marginal transitions to continuous time, we will provide an ablation study to demonstrate how the marginal transitions improve over uniform transitions.
>
> **the lack of an ablation study makes it difficult to determine the extent to which each component contributes**  We acknowledge that assessing the respective contributions of the continuous-time framework and the RRWP encoding requires an ablation study. We will add the ablation in a revised version.
>
> **why is the discrete-time sampling scheme referred to as ancestral sampling** The term ancestral sampling refers to how discrete-time diffusion models are sampled from and was introduced by Song et al. 2021. The important point here is that discrete-time models use a discrete time scale, which is fixed during training, and that allows for only one sampling scheme. Continuous-time models, however, leave the discretization of the time axis to the sampling stage, and offer the possibility to choose any CTMC simulation tool to simulate the reverse process
>
> **How does the continuous-time Markov chain (CTMC) version of discrete-state marginal transition diffusion aid in graph generation?** The first motivation for lifting discrete graph diffusion models to continuous time is that this has proven to be beneficial in terms of performance in several fields, e.g., images (Song et al, 2021) and text (Lou, 2024). Empirically,  we also found that using a continuous-time approach is beneficial for graph generation. Another motivation for lifting graph diffusion models to continuous time is the flexibility it provides in terms of sampling methods. Several of them have been used to sample from continuous-time discrete diffusion models (e.g. Euler, tau-leaping). The design space for such methods is still open, and we believe that our work will open the way to creating innovative sampling schemes for graph diffusion models.
>
> **why is the difference between nodes and edges considered a challenge when the treatment appears to be the same throughout the paper?** The difference in treatment lies in the rate matrices and their associated prior distributions. Since edges and nodes do not have the same set of attributes or support, their corresponding rate matrices have different dimensionality, and so do the probability vectors of their corresponding prior distribution. This differs from, e.g., text data, for which the same rate matrix can be used for every token.
>
> **The model performs well on QM9 but not as well on the other two molecular datasets.**
> We believe this is a misinterpretation of the results on MOSES and GuacaMol. If QM9 is now more a sanity check than a real benchmark, MOSES and GuacaMol are still challenging datasets for general purpose graph generation models.
> If you compare Cometh's results to those obtained by DiGress, you’ll see a neat improvement in performance, especially on GuacaMol. Consequently, Cometh outperforms its discrete-time counterpart and performs very well as a graph generation model.
> We believe your question comes from the comparison to other baselines. Excluding DiGress, all other baselines are trained using domain-specific knowledge (e.g., process SMILES strings), which gives them a clear advantage over general-purpose graph diffusion models. We’d like to highlight the fact that Cometh is the first general-purpose graph diffusion model that obtains near-SOTA performance on GuacaMol, being only outperformed in terms of FCD by the LSTM model.
>
> If you are satisfied with our responses, please consider adjusting your score. Thank you.

---

> > ### Comment · Reviewer_5bR2 · 2024-12-03
> > **Response to the authors**
> >
> > I appreciate the effort,  I will keep the original assessment.

---

### Official Review · Reviewer_Lmdy · 2024-11-03

**Soundness:** 3
**Presentation:** 2
**Contribution:** 2
**Rating:** 5
**Confidence:** 4

**Summary:**

This paper presents Cometh, a continuous-time discrete-state diffusion model for graph generation that combines the flexibility of continuous time with the structure-aware benefits of discrete-state modeling. The authors replace prior structural encodings with a simple random-walk-based encoding to enhance model expressiveness. Experiments show that Cometh achieves state-of-the-art performance across multiple benchmarks, notably outperforming existing models like DiGress on the GuacaMol dataset.

**Strengths:**

1. The discrete-state continuous-time framework for graph generation offers flexibility and is well-suited to the structural properties of graphs.
2. The authors conducted extensive experiments across datasets of varying scales, performed ablation studies, and presented results for conditional generation.

**Weaknesses:**

1. In lines 70-71, the authors claim that this is the first work using a continuous-time discrete-state diffusion model for graph generation. However, this is inaccurate as there is a concurrent work, [R1] (Xu et al., 2024). While a direct comparison is not required for Arxiv paper, it would enhance the paper's completeness to mention this work in the submission.

[R1] Xu, Zhe, et al. "Discrete-state Continuous-time Diffusion for Graph Generation." arXiv preprint arXiv:2405.11416 (2024).

2. The writing and organization of this paper need improvement. There is no whole picture of the problem and the proposed model. In the method section, there are many statements like “we followed Vignac et al. (2022)” and “Following the design choice of Campbell et al. (2022).” Although the authors improve upon previous methods, it would be clearer to distinguish the proposed method from prior work using a summary table or a separate preliminary section.

3. Some experimental results lack sufficient interpretation. For instance, in Table 1, DiGress appears to outperform the proposed COMETH in several metrics (Degree and Orbit), yet no explanation is provided. Additionally, the performance variation between COMETH and COMETH-PC for the Orbit metric is unexplained. Furthermore, the COMETH-PC results are omitted from Table 1 (SBM) and Table 2. Although the authors state that COMETH-PC does not improve SBM performance, including these results would contribute to completeness.

**Questions:**

1. In lines 130-131, the forward process starts with the marginal distribution, whereas in most diffusion models, it starts from a (discrete) uniform distribution. What is the reason for choosing the marginal distribution?
2. The computational resources and time requirements are not reported in the submission.

---

> ### Author Response · Authors · 2024-11-21
>
> We thank the reviewer for the questions and the valuable suggestions to improve the clarity of our work.
>
> **there is a concurrent work, [R1] (Xu et al., 2024)** We acknowledge that concurrent work was released at approximately the same time as ours. We will mention them in a revised version of the manuscript and add them to the baselines. Here, we offer a concise comparison with their work :
>
> - On MOSES, Cometh performs on par with Disco, our competitor’s model. Disco slightly outperforms Cometh regarding VUN, but Cometh obtains better results on all the other metrics.
> - On GuacaMol, Cometh outperforms Disco by a large margin on all the metrics.
> - The results are more contrasted on the synthetic datasets. Disco obtains better results on the distribution metrics (Degree, Clustering, Orbit, and Spectrum), but Cometh obtains much better results regarding Valid, Unique, and Novel samples. Note that the distribution metrics are calculated on all the generated graphs, not only the Valid, Unique, and Novel ones. Therefore, it seems that our superior performance on the VUN metric comes at the cost of a slightly worse ability to retrieve the graph statistics and vice versa.
>
> Generally speaking, we believe that our work provides more significant contributions than Xu et al, 2024 in the sense that their work consists mostly of an extension of DiGress to continuous time. That is, their major contribution is to propose to use of an MPNN rather than a GT as backbone, but that approach yields quite poor experimental results. On the other hand, we successfully leverage the RRWP encoding and the predictor-corrector mechanism and propose a conditional generation experiment.
>
> **The writing and organization of this paper need improvement.** We acknowledge the fact that we refer several times to Vignac et al. (2022) and Campbell et al. (2022) because Cometh relies a lot on the concepts introduced in those two articles. In the final version, we will highlight our contributions better. We also provided a complete description of Campbell et al. 2022 in the Appendix.
> Note that an overview of our model is provided in Figure 1.
>
> **Some experimental results lack sufficient interpretation.** As these are quite complicated architectures, like any deep learning architecture, it is hard to interpret the results for each metric individually. As you can see, DiGress performs better on Degree and Orbit but much worse on Cluster. Overall, we believe that Cometh obtains a much stronger performance than DiGress since we obtain a much better performance on the VUN metric and a competitive performance on most distribution metrics.
>
> **What is the reason for choosing the marginal distribution?** Using marginal transitions along with a marginal prior distribution is a well-established standard in the graph diffusion model literature (Vignac et al. 2022, Qin et al. 2023). We elaborate on this in l 202. To provide further clarification on that matter, the marginal prior distribution is optimal within the space of distributions we are seeking to use, i.e. a distribution that is the product of the same distribution for the n nodes and the n(n-1)/2 edges. This is the closest distribution of that form from the data distribution. A major flaw of the uniform distribution in the case of graphs is that G^{T} would be much denser than the graphs of the data distribution, e.g., for Planar, it would yield graphs of density 0.5 while the average density of this distribution is close to 0.9. Therefore, the model would first need to retrieve this sparsity before fine-tuning the graph structure.
>
> **The computational resources and time requirements are not reported in the submission.** Computation resources are listed in Appendix C5. We will add the time requirements in a revised version.
>
> If you are satisfied with our responses, please consider adjusting your score. Thank you.

---

> > ### Comment · Reviewer_Lmdy · 2024-12-02
> >
> > Thank you for your responses. While some of my concerns have been addressed, I will keep my score unchanged as the limitations pointed out by other reviewers may still exist.

---

### Author Response · Authors · 2024-11-21
**Answer to all reviewers**

We thank all the reviewers for their insightful comments and constructive feedback. We acknowledge that some of our design choices require additional experimental evidence. To address this, we are conducting several ablation studies. Furthermore, we will provide the code to ensure the reproducibility of our experiments.

We will also upload a revised version of the manuscript that incorporates all the comments and suggestions from the reviewers. In the meantime, we are happy to address any remaining questions about our work and provide further clarification if needs be.

---

### Author Response · Authors · 2024-11-25
**Revised manuscript and ablation studies**

We thank again all the reviewers for the valuable feedbacks they provided. We have uploaded a revised version of the manuscript incorporating your suggestions.

Here, we summarize the additional ablations added in Appendix D:
- Ablation on the noise model: We conducted experiments using the uniform noise model in place of the marginal noise model.
- Ablation on the positional encoding: The RRWP encoding was replaced with the feature set used in DiGress for comparison.

Our results demonstrate that marginal transitions combined with RRWP encoding consistently improve performance, particularly in terms of VUN on synthetic datasets.

- Ablation on the loss function: We provided empirical evidence showing the inefficiency of the ELBO as an optimization objective. Specifically, validation performance curves reveal that cross-entropy quickly achieves near-perfect performance, whereas the ELBO saturates below 80%.
- Ablation on the noise schedule: Following reviewer hvvy's suggestion, we conducted a small experiment on QM9 using the exponential noise schedule introduced in Campbell et al., 2022, for categorical settings.

Additionally, we reorganized the paper for improved readability, turning parts of the method section into subsections. The paragraph explaining the tau-leaping procedure was moved to the appendix, as it is not specific to graph generation. Finally, we are also adding our code as supplementary material.

We are happy to answer any remaining questions.

---

### Meta-Review · Area_Chair_K1gv · 2024-12-14

**Metareview:**

**Summary**: This paper proposes a continuous-time discrete-state diffusion model for graphs, dubbed Cometh. Cometh combines the flexibility of continuous time with the structure-aware benefits of discrete-state modeling, with replacing structural encodings to a simple random-walk-based encoding to enhance model expressiveness.

**Strengths and weakness**: Generally, this paper has clarity of presentation and rich experimental results. The proposed discrete-state continuous-time framework for graph generation offers flexibility and is well-suited to the structural properties of graphs. However, reviewers raise the issues about paper organization,  technical novelty and interpretation of results. Overall, all reviewers trend to the rejection of this paper.  Therefore, I cannot recommend accepting the paper at this time.

 For the improvement of the paper's quality, it's better to add more discussion between this paper and previous continuous-time diffusion models.

**Additional Comments On Reviewer Discussion:**

During the discussion time, the authors respond the reviewer's concerns. All the reviewers acknowledge reviewers' rebuttals and decide to keep the original score.

---

### Decision · Program_Chairs · 2025-01-22

Reject